# Enhancing Diffusion-Based Sampling with Molecular Collective Variables

**Juno Nam**[2,*,†]   **Bálint Máté**[3,†]   **Artur P. Toshev**[4,†]   **Manasa Kaniselvan**[5,†]
**Rafael Gómez-Bombarelli**[2]   **Ricky T. Q. Chen**[1]   **Brandon Wood**[1]
**Guan-Horng Liu**[1,*]   **Benjamin Kurt Miller**[1,*]

[1]FAIR at Meta   [2]MIT   [3]University of Geneva   [4]TUM   [5]ETH Zurich
[*]Core contributors   [†]Work done during internship at FAIR

## Abstract

Diffusion-based samplers learn to sample complex, high-dimensional distributions using energies or log densities alone, without training data. Yet, they remain impractical for molecular sampling because they are often slower than molecular dynamics and miss thermodynamically relevant modes. Inspired by enhanced sampling, we encourage exploration by introducing a sequential bias along bespoke, information-rich, low-dimensional projections of atomic coordinates known as collective variables (CVs). We introduce a repulsive potential centered on the CVs from recent samples, which pushes future samples towards novel CV regions and effectively increases the temperature in the projected space. Our resulting method improves efficiency, mode discovery, enables the estimation of free energy differences, and retains independent sampling from the approximate Boltzmann distribution via reweighting by the bias. On standard peptide conformational sampling benchmarks, the method recovers diverse conformational states and accurate free energy profiles. We are the first to demonstrate reactive sampling using a diffusion-based sampler, capturing bond breaking and formation with universal interatomic potentials at near-first-principles accuracy. The approach resolves reactive energy landscapes at a fraction of the wall-clock time of standard sampling methods, advancing diffusion-based sampling towards practical use in molecular sciences.

## 1 Introduction

A central goal of statistical mechanics simulations is to estimate macroscopic thermodynamic properties, such as the energy released in a chemical transformation. These simulations typically employ molecular dynamics (MD) or Markov chain Monte Carlo (MCMC), which propagate an atomic structure forward in time or along a chain using energies from quantum mechanical calculations or their approximations (Marx & Hutter, 2000; Frenkel & Smit, 2023; Jacobs et al., 2025). Over long times, the distribution of states converges to the Boltzmann distribution $\nu(x) \propto \exp(-u(x))$, where the reduced energy $u$ encodes the chemical system, temperature, and other thermodynamic variables. Sampling from this distribution is deceptively difficult, despite having access to its explicit unnormalized form, due to the large number of possible atomic configurations and the high energy barriers separating them. Straightforward MD or MCMC often requires an impractically large number of sequential energy evaluations to capture conformational changes or chemical reactions.

Machine learning (ML)-based samplers attempt to bypass dynamics by reformulating sampling as statistical inference on the Boltzmann distribution, using its log-likelihood to draw independent and identically distributed (i.i.d.) samples. Generally, models that approximate thermodynamic ensembles are known as *Boltzmann Generators*. The first work in this area applied normalizing flows (Noé et al., 2019), while recent approaches employ diffusion-based samplers that steer stochastic differential equations (SDEs) whose trajectories yield samples consistent with the target distribution (Zhang & Chen, 2021; Vargas et al., 2023a; Havens et al., 2025).[1] Although training can amortize

---

[1]More information about neural and diffusion-based samplers can be found in Section A.2.

effort across multiple draws, these methods do not inherently reduce the number of energy evaluations required to obtain accurate equilibrium statistics compared to MD-based approaches (He et al., 2025). Directly applying diffusion samplers faces a well-known pitfall of mode collapse, where training and sampling concentrate on high-probability basins while underrepresenting rare but thermodynamically important states. This issue is not only about exploration but also about statistics, since reliable free energies and ensemble averages depend critically on assigning correct weights to rare configurations, which may be exponentially small. Thus, successful sampling must both reach low-occupancy modes and attribute the correct equilibrium weights to them.

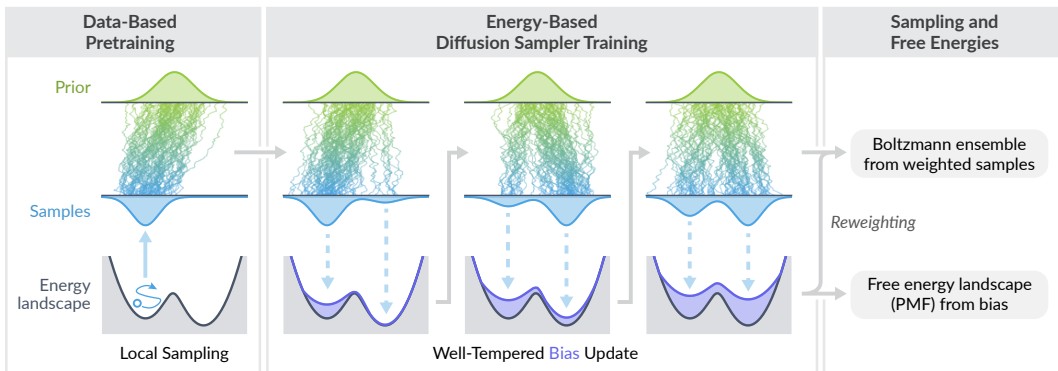

Figure 1: **Scheme for neural enhanced sampling.** (Left) Local sampling and pretraining near the reference configuration. (Middle) Energy-based sampler training with a CV-space bias that is updated online via well-tempered deposition. (Right) After convergence, the final bias yields the potential of mean force (PMF) along CVs; also, reweighting recovers the Boltzmann ensemble, including samples and free energy differences.

Inspired by enhanced sampling methods, we augment the diffusion-based sampler with a bias in collective variable (CV) space. CVs are low-dimensional functions of atomic coordinates that capture slow, chemically relevant motions and can be chosen from common system characteristics (e.g., bond lengths or torsional angles) or identified with ML (Sidky et al., 2020; Bonati et al., 2023). During the training of a diffusion-based sampler, the sampler iteratively proposes configurations and receives feedback from their energies. An additional online repulsive potential is maintained over the CVs: regions visited more often accumulate higher bias, which raises their effective energy, discouraging repeated visits. This flattens free energy barriers over training progress and mitigates collapse onto dominant modes. At inference, importance weights remove the effect of the bias, ensuring that ensemble estimates match the target Boltzmann distribution. The result is broader exploration, including rare modes, while preserving statistically correct estimates after reweighting. Furthermore, compared to MD-based enhanced sampling, the diffusion process mixes more efficiently within the biased distribution, enabling faster exploration. Our contributions are as follows:

- We propose the *Well-Tempered Adjoint Schrödinger Bridge Sampler* (WT-ASBS), which augments the state-of-the-art diffusion-based sampler ASBS (Liu et al., 2025) with a biasing mechanism inspired by well-tempered metadynamics (WTMetaD, Barducci et al., 2008), achieving substantially improved exploration and accurate estimation of free energy differences.

- We develop a protocol for applying WT-ASBS to molecular systems, demonstrating (i) accurate sampling of peptide conformations in Cartesian coordinates, where the baseline method fails; and (ii) the first diffusion-based sampling of reactive landscapes, achieving shorter wall-clock times than WTMetaD. This establishes a new practical role for diffusion-based samplers in chemistry.

## 2 BACKGROUND

### 2.1 MOLECULAR SAMPLING

**Problem setup** In the configurational space $\mathcal{X} \subseteq \mathbb{R}^{n \times 3}$ for $n$ atoms with elemental identities $a \in \mathcal{E}^n$ with a set of element types $\mathcal{E}$, the potential energy function $E : \mathcal{X} \times \mathcal{E}^n \to \mathbb{R}$ defines the Boltzmann distribution at inverse temperature $\beta = 1/k_{\mathrm{B}}T$ as $\nu(x; a) \propto \exp(-\beta E(x; a))$. In molecular simulations, the central challenge is to generate configurations representative of the underlying

free energy landscape that describes the relative stability and likelihood of different molecular configurations. Three main downstream applications motivate this task: (i) obtaining a diverse and statistically meaningful set of configurations for structural analysis, (ii) accurately computing free energy differences between relevant states, and (iii) uncovering transition mechanisms that govern rare events that are nevertheless thermodynamically essential. For notational convenience, we omit elemental identities after this paragraph and write $E(x)$ and $\nu(x)$.

One rarely aims to sample the entirety of $\mathcal{X}$ in practice because it includes *all* configurations consistent with the chosen composition and potential energy function. Instead, sampling is naturally restricted to an accessible component $\mathcal{A} \subset \mathcal{X}$ that is kinetically connected to a reference conformer $x_{\text{ref}} \in \mathcal{X}$ on the timescales of interest at temperature $T$ by physics. In molecular conformational sampling, $\mathcal{A}$ implies configurations that preserve the bond topology of $x_{\text{ref}}$. If one included bond reorganizations occurring on longer timescales (hours to days), $\mathcal{A}$ would have to grow to include these states. This restriction ensures that the ensemble reflects the connected set of metastable states and transition pathways relevant to the molecular process of interest, while excluding kinetically inaccessible or chemically irrelevant configurations, such as incorrect molecular chirality.

While basic sampling algorithms aim to produce i.i.d. samples from $\nu(x)$, the exponential dependence of probabilities on $E(x)$ leads to significant undersampling of low-population modes even at moderate energy differences. This is especially important when high-energy transition configurations are being studied. Consequently, many effective schemes for molecular sampling generate weighted samples $\{(X_i, W_i)\}_{i=1}^{N}$ with $X_i \in \mathcal{A}$ and $W_i \geq 0$, obtained for example via biased simulations with reweighting. For any bounded observable $f : \mathcal{X} \to \mathbb{R}$, the self-normalized estimator

$$\hat{\nu}_n(f) = \frac{\sum_{i=1}^{N} W_i f(X_i)}{\sum_{i=1}^{N} W_i} \xrightarrow{N \to \infty} \mathbb{E}_\nu[f(X) \mid X \in \mathcal{A}] \tag{1}$$

recovers expectations under the Boltzmann ensemble restricted to $\mathcal{A}$. We now summarize the requirements and discuss the practical realization of these:

1. **Restricted Support:** Sampling is confined to the relevant metastable state and transitions. Configurations in $\mathcal{X} \setminus \mathcal{A}$ are not produced or receive zero weight.

2. **Consistency:** The reweighted ensemble converges to the Boltzmann ensemble on $\mathcal{A}$, i.e., convergence in Eq. (1) holds for all bounded $f$.

3. **Local Efficiency:** Within any chemically relevant region of $\mathcal{A}$, the procedure yields sufficient sample diversity to estimate observables and capture transition mechanisms.

**Requirement 1: Restrained sampling** Membership in $\mathcal{A}$ is governed by the free energy barriers that separate modes, not by the stability (state free energies) of individual modes. When sampling is performed by emulating physical dynamics (e.g., MD), high barriers suppress transitions and trajectories remain within $\mathcal{A}$ by construction. For sampling procedures that do not generate time-correlated frames, one should enforce the restriction explicitly using restraint potentials that are flat on $\mathcal{A}$ while strongly penalizing $\mathcal{X} \setminus \mathcal{A}$. This confines exploration to the intended modes while allowing efficient coverage within them. Practical considerations are further discussed in Section 3.2.

**Requirements 2 and 3: Biasing on the CV space** High free energy barriers often align with a small set of slow coordinates, so unbiased exploration in $\mathcal{A}$ can waste effort on fast motions while undersampling rare but important transitions. CVs address this by mapping configurations to a low-dimensional space $\xi : \mathcal{X} \to \mathcal{S} \subset \mathbb{R}^m$ with $m \ll n$, that capture functionally relevant degrees of freedom. Examples include peptide backbone torsions $(\phi, \psi)$ and the center-of-mass distance between a protein pocket and its ligand. By steering sampling with a bias that depends only on $s = \xi(x)$, we enhance explorations across modes while leaving orthogonal fluctuations unchanged. A bias $V(s)$ yields the unnormalized sampling density $\nu_V(x)$ and importance weights $w(x)$ as

$$\nu_V(x) \propto \exp[-\beta E(x) - \beta V(\xi(x))], \quad (2) \qquad w(x) \propto \frac{\nu(x)}{\nu_V(x)} \propto \exp[+\beta V(\xi(x))], \quad (3)$$

so unbiased expectations are recovered by the estimator in Eq. (1) with $X_i \sim \nu_V$ and $W_i = w(X_i)$, ensuring Requirement 2.

Biasing-based enhanced sampling approaches often construct a bias that raises the effective sampling temperature along CV space, introduced by Barducci et al. (2008) as the *well-tempered* target

distribution. We first define the CV marginal $\bar{\nu}(s)$ and the potential of mean force (PMF) $F(s)$ from the unbiased CV marginal (defined up to a constant)[2] using the Dirac delta function $\delta$:

$$\bar{\nu}_V(s) \propto \int_{\mathcal{X}} \nu_V(x)\delta[\xi(x) - s]\,\mathrm{d}x, \qquad (4) \qquad\qquad F(s) := -\tfrac{1}{\beta}\log\bar{\nu}(s). \qquad (5)$$

Now, given a bias factor $\gamma > 1$ that acts as a scale factor for $T$, we define the well-tempered bias $V_{\mathrm{WT}}(s)$ and the corresponding target distribution $\nu_{\mathrm{WT}}(x)$ from Eq. (2) as

$$V_{\mathrm{WT}}(s) = -(1 - \tfrac{1}{\gamma})F(s), \qquad (6) \qquad \nu_{\mathrm{WT}}(x) \propto \exp[-\beta E(x) + \beta(1 - \tfrac{1}{\gamma})F(\xi(x))], \qquad (7)$$

From Eqs. (4), (5) and (7), the well-tempered CV marginal is $\bar{\nu}_{\mathrm{WT}}(s) \propto \exp[-\tfrac{\beta}{\gamma}F(s)] = [\bar{\nu}(s)]^{1/\gamma}$, which shows that the sampled CV behaves as if at a higher effective temperature $T_{\mathrm{eff}} = \gamma T$ ($\beta_{\mathrm{eff}} = \beta/\gamma$), while the orthogonal conditional remains unchanged: $\nu_{\mathrm{WT}}(x\,|\,\xi(x) = s) = \nu(x\,|\,\xi(x) = s)$. This selective increase in sampling temperature along the CV satisfies Requirement 3. To construct $V_{\mathrm{WT}}$ on-the-fly during simulation, one can stack Gaussian kernels (Barducci et al., 2008), use a kernel density estimator (Invernizzi & Parrinello, 2020), or obtain the bias variationally (Valsson & Parrinello, 2014; Bonati et al., 2019). These requirements also extends to a variety of molecular enhanced sampling algorithms, as discussed in Section A.1.

**Mode exploration** Requirement 2 is sometimes relaxed so that we generate samples uniformly across modes without weighting. This is a practically relevant setting that emphasizes capturing the diversity of metastable modes rather than reproducing their exact Boltzmann probabilities. Mode exploration is the usual objective of data-driven generative models for molecular conformations (Jing et al., 2022), protein structures (Watson et al., 2023), and crystalline materials (Jiao et al., 2023; Miller et al., 2024), which are trained to approximate the empirical distribution of representative structures extracted from exploratory simulations or experimental data without Boltzmann weights.

## 2.2 DIFFUSION-BASED NEURAL SAMPLERS

**Neural samplers** Diffusion-based neural samplers learn a control vector field that steers a stochastic process to gradually transform noise into a target distribution, typically specified by an unnormalized density or energy function. Unlike iterative generative methods trained on data from implicitly defined distributions (Sohl-Dickstein et al., 2015; Ho et al., 2020; Lipman et al., 2022; Albergo et al., 2023), neural samplers can generate samples by querying the energy alone. This ability is especially relevant for Boltzmann distributions (Zhang & Chen, 2021; Berner et al., 2022; Vargas et al., 2023a; Havens et al., 2025). More information about neural samplers is provided in Section A.2.

**Adjoint Schrödinger Bridge Sampler (ASBS, Liu et al., 2025)** ASBS casts Boltzmann sampling as the Schrödinger Bridge (SB) problem between source $\mu(X_0)$ and Boltzmann target $\nu(X_1) \propto \exp(-\beta E(X_1))$ that seeks the optimal control $u_\theta$ w.r.t. the path Kullback-Leibler (KL) objective

$$\min_u D_{\mathrm{KL}}(p^u \| p^{\mathrm{base}} = p^{u=0}) = \mathbb{E}_{X \sim p^u}\left[\int_0^1 \tfrac{1}{2}\|u_\theta(X_t, t)\|^2\,\mathrm{d}t\right], \qquad (8)$$

$$\text{s.t. } \mathrm{d}X_t = \sigma_t u_\theta(X_t, t)\,\mathrm{d}t + \sigma_t\,\mathrm{d}W_t, \; X_0 \sim \mu(X_0), \; X_1 \sim \nu(X_1), \qquad (9)$$

where $\sigma_t$ is the noise schedule, and $p^u$ denotes the law of the SDE in Eq. (9).[3] ASBS avoids backpropagating through the SDE during optimization by introducing the parametrized corrector $h_\phi(X_1)$ that debiases the effect of endpoint coupling from base distribution to allow the SOC interpretation of Eq. (8), which can be converted to Adjoint Matching (AM) objective for the drift (Havens et al., 2025). The corrector is then learned by optimizing the Corrector Matching (CM) objective:

$$\text{(AM)} \quad u = \arg\min_u \mathbb{E}_{p^{\mathrm{base}}_{t|0,1} p^{\bar{u}}_{0,1}}[\|u_\theta(X_t, t) + \sigma_t(\beta\nabla E(X_1) + h_\phi(X_1))\|^2], \qquad (10)$$

$$\text{(CM)} \quad h = \arg\min_h \mathbb{E}_{p^{\bar{u}}_{0,1}}[\|h_\phi(X_1) - \nabla_{X_1}\log p^{\mathrm{base}}(X_1|X_0)\|^2], \qquad (11)$$

where $\bar{u} := \texttt{stopgrad}(u)$. These paired objectives instantiate iterative proportional fitting (IPF, Kullback, 1968) and converge to the SB solution. The expectations use closed-form base conditionals, so no target samples or importance weights are needed, which preserves scalability. Furthermore, the control model supports approximate data-based pretraining (see Section C).

---

[2]The term PMF for $F(s)$ is used interchangeably with the *free energy profile/landscape* along the CV.

[3]While the original ASBS formulation is based on SDEs with general drift $f(X_t, t) + \sigma_t u_\theta(X_t, t)$, here we only consider the driftless base process with $f = 0$.

ASBS achieved strong performance on synthetic potentials and amortized conformer sampling (*mode exploration* setting) compared to previous diffusion-based samplers. However, it missed less populated modes in alanine dipeptide sampling due to its mode-seeking behavior, motivating the introduction of an enhanced sampling mechanism in this work.

## 3 METHODS

Now we augment ASBS with a well-tempered bias on a small set of CVs, updated online from i.i.d. model samples during training, instantiating a *neural enhanced sampler* that satisfies the requirements in Section 2.1. We first present the WT-ASBS setup and mechanism together with a convergence guarantee. We then detail practical strategies with data-based pretraining, WT-ASBS training, and post-training reweighting to recover the Boltzmann ensemble (scheme in Fig. 1).

### 3.1 WELL-TEMPERED ADJOINT SCHRÖDINGER BRIDGE SAMPLER

---

**Algorithm 1** Well-Tempered Adjoint Schrödinger Bridge Sampler (WT-ASBS)

---

**Require:** prior $\mu$, potential energy $E(x)$, CV $\xi(x)$, control $u_\theta(x,t)$, corrector $h_\phi(x)$
1: Initialize $V_0(s) \leftarrow 0$
2: **for** Outer step $k = 0, 1, \cdots$ **do**
3:      **for** Inner step $l = 0, 1, \cdots$ **do**
4:          $\theta \leftarrow \arg\min_\theta \mathbb{E}_{p_{t|0,1}^{\text{base}} p_{0,1}^{\bar{u}}}[\|u_\theta(X_t,t) + \sigma_t(\beta\nabla(E + V_k \circ \xi) + h_\phi)(X_1)\|^2]$    ▷ *Adjoint Matching*
5:          $\phi \leftarrow \arg\min_\phi \mathbb{E}_{p_{0,1}^{\bar{u}}}[\|h_\phi(X_1) - \nabla_{X_1} \log p^{\text{base}}(X_1|X_0)\|^2]$              ▷ *Corrector Matching*
6:      Sample i.i.d. $\{X_{1,k}^{(i)}\}_{i=1}^N \sim p_1^u$,    $s_k^{(i)} \leftarrow \xi(X_{1,k}^{(i)})$
7:      $V_{k+1}(s) \leftarrow V_k(s) + h\sum_{i=1}^N \exp(-\frac{\beta}{\gamma-1}V_k(s_k^{(i)}))\exp(-\frac{\|s-s_k^{(i)}\|^2}{2\sigma^2})$    ▷ *Well-Tempered Bias Update*
8: **output** Final control $u_\theta^*(x,t)$, final bias $V^*(s)$

---

**Setup and two-time-scale algorithm** We implement WT-ASBS with a well-tempered bias update through stacking Gaussian kernels (Barducci et al., 2008) to reach the well-tempered target Eq. (6). Let $\mathcal{S} \subset \mathbb{R}^m$ denote a compact CV region of interest, where bias potentials $V : \mathcal{S} \to \mathbb{R}$ are defined and constructed, which induces the sampling distribution $\nu_V(x) \propto \exp[-\beta(E(x) + V(\xi(x)))]$. WT-ASBS presented in Algorithm 1 is executed via a two-time-scale scheme:

1. *Inner step (ASBS training).* For fixed $V_k$, train ASBS with energy $E + V_k \circ \xi$ until convergence so that the marginal distribution at $t = 1$ equals $\nu_{V_k}$.
2. *Outer step (well-tempered bias update).* Draw a conditionally i.i.d. batch of configurations from ASBS $\{X_{1,k}^{(i)}\}_{i=1}^N \sim p_1^u$, project to CV space $s_k^{(i)} = \xi(X_{1,k}^{(i)})$, and update

$$V_{k+1}(s) = V_k(s) + h\sum_{i=1}^N \exp\big(-\tfrac{\beta}{\gamma-1}V_k(s_k^{(i)})\big)K_\sigma(s, s_k^{(i)}), \tag{12}$$

with fixed height $h > 0$ and Gaussian kernel $K_\sigma(s, s') = \exp(-\frac{\|s-s'\|^2}{2\sigma^2})$.

We further establish a convergence guarantee showing that WT-ASBS provably approaches the desired well-tempered distribution under the two-time-scale update scheme (proof in Section B.1):

**Proposition 3.1** (Convergence of WT-ASBS). *When Algorithm 1 is performed until convergence, the bias potential $V_k$ converges almost surely to $V^*(s) = -(1 - \frac{1}{\gamma})F(s) + \text{const}$, and hence the sampled distribution converges to the well-tempered target Eq. (7).*

*Remark* 3.1. Proposition 3.1 ensures that, upon completion of training, the PMF along the CVs can be recovered up to a constant from the final bias as $F(s) = -\frac{\gamma}{\gamma-1}V^*(s) + \text{const}$.

**Practical implementation** In practice, AM uses a replay buffer: gradient signals are not computed directly from samples of $p_{0,1}^{\bar{u}}$ with the current $u_\theta$, but rather from subsets drawn from stored samples in the buffer. Assuming the bias deposition per step is small and AM training is efficient enough for generated samples to follow the current bias closely, we can *merge* the outer and inner sample generation steps by updating the bias during AM training. See Section C for details.

Although our biasing mechanism mirrors classical MD-based enhanced sampling, neural samplers like ASBS generate i.i.d. samples from the current distribution, eliminating the need to wait for decorrelation before bias deposition occurs. This allows us to exploit the bias deposition mechanism more efficiently to explore the regions of configurational space. We further discuss recent works that promote exploration by shaping the annealing path or by modifying the learning objective in Section A.2.

## 3.2 Recipe for Efficient Sampling of Molecular Systems

Here, we outline three practical ingredients for deploying WT-ASBS to molecular systems.

**Warm-start with localized pretraining** In this work, we consider a single reference configuration that defines the accessible component $\mathcal{A}$, corresponding to the initial conformer used to start MD or MCMC chains. Although Algorithm 1 does not require data samples for training, as in the original ASBS, the model can be pretrained with available samples to warm-start and improve training efficiency. Accordingly, we perform short MD simulations (local sampling) to generate samples from the mode containing the reference configuration and apply bridge matching pretraining (Section C) to initialize the control model. During early-stage local conformational exploration without energetic barriers, correlated samplers such as MD or MCMC are more efficient because they move locally on the energy landscape, unlike diffusion samplers that must transform noise to physical samples through hundreds of model evaluations for every sample. Since this is only a pretraining stage, high accuracy is unnecessary, and we can further reduce computational cost by using approximate energy evaluations (e.g., classical force fields), faster sampling settings, or transfer operator surrogates (e.g., Schreiner et al., 2023) to create the pretraining data.

**WT-ASBS training with CVs and restraints** Upon adopting ASBS, the algorithm introduces several additional hyperparameters governing training. The first is the choice of low-dimensional CVs that defines the space over which modes are explored. CVs may be selected either from chemical intuition or from topological changes induced by the underlying transformation, as demonstrated in Section 4. For more complex systems, CVs can also be parameterized by shallow neural networks and trained from available conformations (Sidky et al., 2020; Bonati et al., 2023). Other hyperparameters follow conventions from prior enhanced sampling studies. However, the improved mode-mixing behavior of diffusion samplers permits the use of smaller $\gamma$ values, i.e., the effective temperature along the CVs can be lower. Similarly, the Gaussian height $h$ must be reduced, since diffusion samplers generate uncorrelated samples and thus deposit bias more frequently.

To satisfy Requirement 1, we use restraint potentials. The specific combination of restraints is system dependent, but their application can be formalized through the lens of chemical isomerism. Since which types of isomerism are suppressed under physical conditions is well understood, these serve as a natural guide (Canfield et al., 2018). We provide a detailed treatment in Section A.4, and here illustrate the idea with chirality inversion in proteins and peptides. The $C_\alpha$ atom of each amino acid is a chiral center: while both L- and D-configurations exist, nearly all natural amino acids only adopt the L-configuration. As the energy function does not inherently distinguish mirror-symmetric configurations, we introduce flat-bottom harmonic potentials on improper torsional angles defining chirality, thereby biasing sampling toward the physically relevant L-configurations.

**Sampling and refinement** Remark 3.1 implies that the PMF along the CVs can be recovered directly from the final bias. While samples stored in the buffer may be analyzed to characterize configurations associated with specific regions in CV space, to recover the full weighted conformational distribution, one can integrate Eq. (9) to generate samples $X_i$ and assign weights $W_i = \exp[\beta V^*(\xi(X_i))]$, yielding the self-normalized estimator in Eq. (1). Note that this is exact only in the limit of a perfectly trained sampler. In principle, the path importance weights in ASBS could be used to debias an imperfect sampler with the gradient-based parametrization of control and corrector (Liu et al., 2025), though these weights may suffer from high variance. The learned bias also enables a practical refinement strategy: short MD or MCMC trajectories are initiated from the generated samples under the biased energy $E + V^* \circ \xi$, and the resulting configurations are reweighted with $V^*$. This restores asymptotic correctness while keeping the procedure computationally efficient thanks to the smoother energy landscape along the CVs.

## 4  EXPERIMENTS

We demonstrate the efficiency of the WT-ASBS pipeline on four molecular sampling tasks involving peptides and chemical reactions. Since Requirements 1 and 3 are incorporated into the design of the method, correctness with respect to Requirement 2 ensures that all requirements are satisfied. We therefore assess whether the empirical PMF, $\hat{F}(s) = -\frac{1}{\beta}\log\hat{\nu}(s)$ obtained from reweighted samples Eq. (1) and the PMF from bias (Remark 3.1) agree with those from reference densities. Accuracy is computed from the PMF by measuring the free energy of state $i$ in conformational states as

$$F_i = -\tfrac{1}{\beta}\log\int_{\mathcal{S}}\exp(-\beta F(s))\mathbf{1}[s \in \mathcal{S}_i]\,\mathrm{d}s \,+\, \mathrm{const}, \tag{13}$$

where $\mathcal{S}_i$ is the CV region of state $i$. We report free energy differences $\Delta F_{ij} = F_j - F_i$, thereby removing the additive constant as it is the same for all $\mathcal{S}_i$. We compute multi-state errors with a common offset. Although weighted sampling required by Requirement 3 is best assessed via PMFs and free energies, we additionally compare with diffusion samplers on alanine dipeptide in Section F using standard distributional metrics. Our main comparisons are with ASBS and WTMetaD.

### 4.1  CONFORMATIONAL SAMPLING FOR SMALL PEPTIDES

We first benchmark the WT-ASBS method on two widely studied peptide systems, alanine dipeptide (Ala2; Ace–Ala–Nme) and alanine tetrapeptide (Ala4; Ace–Ala$_3$–Nme), which serve as standard testbeds for evaluating molecular sampling methods. Both peptides are modeled with a classical force field with implicit water solvation, with the chirality restraint applied to all $C_\alpha$ atoms (see Section D). In this work, we focus on sampling the full Cartesian coordinates of the molecules. Although peptides and proteins adopt a broad range of conformations, these arise largely from the combinatorial organization of local backbone torsional states, specifically the $\phi$ and $\psi$ angles (Fig. 2a), as classically illustrated by Ramachandran et al. (1963). Building on this insight, we demonstrate that energy-based neural samplers can be made effective for sampling peptides directly in Cartesian space when augmented with biasing along these torsional angles.

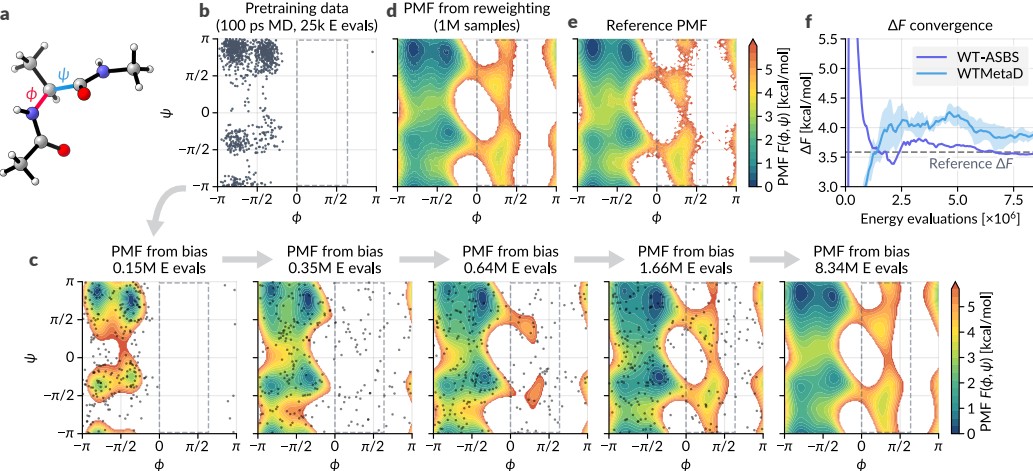

Figure 2: **Sampling alanine dipeptide.** (a) Molecular structure of alanine dipeptide and definition of the torsional collective variables $\phi$ and $\psi$. (b) Distribution of $\phi$ and $\psi$ obtained from a short unbiased MD trajectory used for pretraining the sampler. (c) Evolution of the PMF reconstructed from the accumulated bias during training, illustrating progressive recovery of the major basins in the free energy landscape. The dotted box delineates the two states, one with $\phi < 0$ or $\phi > 2$ and the other with $\phi \in [0, 2]$. (d) PMF estimated by reweighting $10^6$ samples generated by the trained sampler using the final bias. (e) Reference PMF computed from a long unbiased MD simulation for comparison. (f) Convergence of free energy differences $\Delta F$ between two states as a function of number of energy evaluations during training.

**Alanine dipeptide (Ala2)**  We employ $\phi$ and $\psi$ torsion angles as CVs (Fig. 2a). The conformational landscape is broadly divided into two states along $\phi$ (Fig. 2b). Local sampling within the more populated state is efficient due to the absence of a significant barrier; thus, we easily sample

pretraining data from a short MD simulation in this region. The pretrained model produces samples restricted to this state. However, once WT-ASBS training begins, the bias deposition progressively shifts the sampling distribution away from the explored region, enabling the discovery of conformations in the less populated state. Over training progress, the deposition asymptotically converges to the bias potential required to reproduce the PMF (Fig. 2c). Fig. 2d shows the PMF obtained from the generated samples of the fully trained model using weights from the final bias. Both the bias-derived PMF and the reweighted-sample PMF agree closely with the reference PMF (Fig. 2e), whereas the baseline ASBS fails to reproduce the correct distribution in the less populated region (Fig. 14). Moreover, compared to the baseline enhanced sampling method (WTMetaD) with the same bias factor, WT-ASBS achieves convergence of the $\Delta F$ between the two states more accurately.

To quantify the robustness of WT-ASBS and clarify practical choices, we performed ablations of the WT bias hyperparameters and training mechanisms, and reported detailed results in Section E. Varying the Gaussian height $h$, kernel width $\sigma$, and bias factor $\gamma$ over wide ranges leaves the final $\Delta F$ unchanged except at the most extreme $h$, while larger $h$ and broader $\sigma$ mainly accelerate early exploration and barrier crossing. The training mechanism ablations show that overly frequent replay buffer refreshes slow convergence because the sampler does not have time to adapt to the evolving bias, whereas moderate update intervals and localized pretraining lead to a faster approach to the reference $\Delta F$ and reduce the total number of energy evaluations. Furthermore, Section F.4 demonstrates that WT-ASBS can also operate with an automatically learned ML CV (Trizio & Parrinello, 2021), recovering accurate PMF and $\Delta F$ after reweighting, which indicates that the method could still be utilized when suitable low-dimensional CVs are not known a priori.

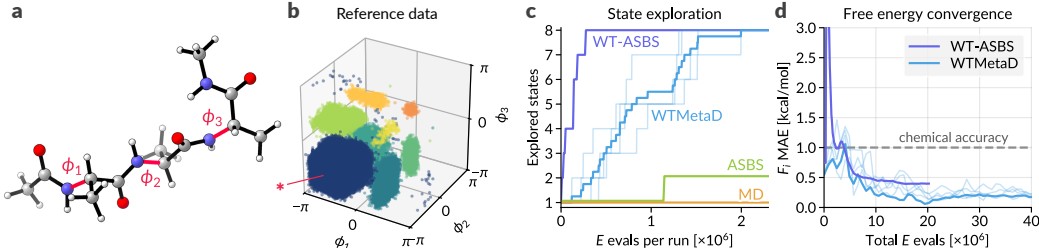

Figure 3: **Sampling alanine tetrapeptide.** (a) Molecular structure of alanine tetrapeptide with torsional collective variables $\phi_1$, $\phi_2$, and $\phi_3$. (b) Reference distribution in torsional space. Each torsion is partitioned into two regions, $\phi_i \in [0, 2]$ and $\phi_i < 0$ or $\phi_i > 2$, producing eight metastable states corresponding to all combinations of these regions across the three $\phi_i$. Pretraining is restricted to the single state indicated by $*$, so the sampler initially encounters only one of the eight possible regions. (c) Accumulated number of explored states during the initial phase of training or simulation (up to 2M energy evaluations), showing how the sampler progressively discovers additional states as the bias is deposited. (d) Convergence of the free energies of the eight states, reported as the MAE up to an additive constant. For WT-ASBS, the curve extends up to 20M energy evaluations where the bias has converged.

**Alanine tetrapeptide (Ala4)**   As seen in the Ala2 case, the main energetic barriers separating conformational states arise from backbone $\phi$ torsions. We therefore employ three CVs, $\phi_1$, $\phi_2$, and $\phi_3$. A long 100-µs reference MD simulation shows that the system occupies eight distinct modes, corresponding to the $2^3$ combinations of the three $\phi$ torsional states (Fig. 3b). We initialize from a single conformation in the indicated mode and generate pretraining data from a short MD simulation confined in the mode (Fig. 15). As shown in Fig. 3c, WT-ASBS explores conformational states far more efficiently than the baseline WTMetaD, discovering all eight modes within the early stages of training. This advantage arises from its ability to generate uncorrelated samples from the biased potential, enabling exploration of multiple directions of conformational change simultaneously, whereas MD-based enhanced sampling explores states sequentially in simulation time. Consequently, WT-ASBS also achieves effective convergence of free energies (Fig. 3d), as measured by the mean absolute error (MAE) of free energies up to a constant $\min_c \frac{1}{8} \sum_i |F_i - F_i^{\text{ref}} + c|$, within chemical accuracy (1 kcal/mol), demonstrating the practical applicability of WT-ASBS. We note that it does not yield lower free energy MAE than WTMetaD, likely because MD already provides efficient local mixing within each basin once barriers along the chosen CV are crossed, whereas the diffusion sampler must learn the full high-dimensional intra-basin distribution. Measure-transport-based global sampling is most beneficial for crossing large barriers, whereas MD-based local sampling approaches

can still perform well when refining physical observables, which further suggests that combining the two strategies may be advantageous for more complex systems.

## 4.2 SAMPLING REACTIVE ENERGY LANDSCAPES

While peptide results highlight the efficiency of WT-ASBS in terms of the number of energy evaluations, the computational budgets in diffusion-based sampling in these cases are dominated by SDE integration with a control network, since classical force fields are extremely fast to evaluate. However, classical force fields are insufficient for accurate conformational sampling, particularly in cases involving bond reorganization, while density functional theory (DFT), the standard for accurate energetics, is computationally prohibitive. Recent advances in semiempirical quantum methods (Froitzheim et al., 2025) and universal ML interatomic potentials (uMLIPs) demonstrate that near-DFT accuracy can be achieved at a fraction of the compute, and open the door to scalable and accurate sampling. To illustrate this direction and compare the computational cost in practical settings, we apply WT-ASBS to sample reactive energy landscapes involving bond formation (Trizio & Parrinello, 2021). We employ a recent uMLIP, UMA-S-1.1 (Wood et al., 2025), which accurately reproduces DFT energies, as the underlying energy model. For reaction sampling, we cross-check with WTMetaD results since sampling the transition region with unbiased samplers like MD is practically impossible (e.g., $\sim 10^{15}$ samples required for a 20 kcal/mol barrier at 300 K).

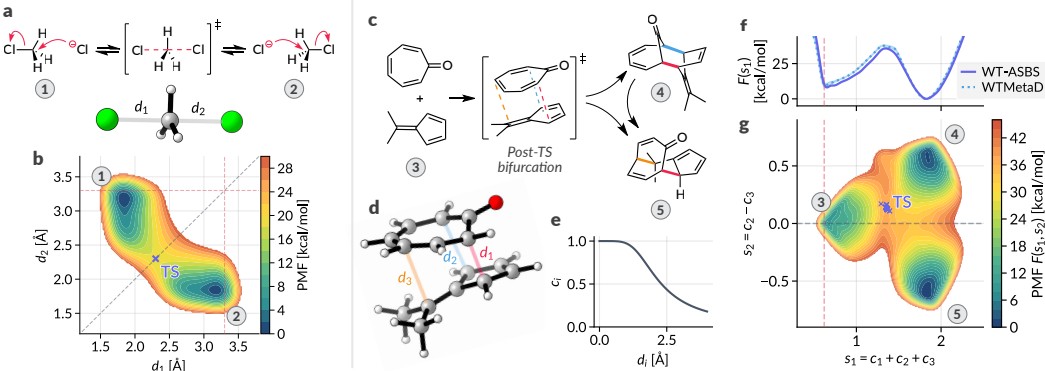

Figure 4: **Sampling reactive landscapes.** (a) $S_N2$ reaction scheme with nucleophilic substitution between states 1 and 2, along with the two bond distance CVs $d_1$ and $d_2$ that describe approach and departure of the chloride ions. (b) PMF $F(d_1, d_2)$ for the $S_N2$ system, showing the reactant and product basins and the transition-state region along the two-dimensional distance coordinates. (c) Cycloaddition reaction scheme exhibiting a post-TS bifurcation, highlighting the competing pathways that originate from the same transition structure. (d) Definition of the three forming bond distances $d_1$, $d_2$, and $d_3$ used to parameterize progress along the bifurcating reaction pathways. (e) Contact CVs $c_i$ obtained from the distances $d_i$, used to construct smoother CVs for sampling. (f) One-dimensional PMF $F(s_1)$ along the CV $s_1 = c_1 + c_2 + c_3$, comparing WT-ASBS and WTMetaD. (g) Two-dimensional PMF $F(s_1, s_2)$ with $s_2 = c_2 - c_3$, resolving the post-TS bifurcation and showing the distinct product channels, with TS locations from saddle point optimization.

**Nucleophilic substitution ($S_N2$)** We study a prototypical $S_N2$ reaction (Fig. 4a), where a chloride ion attacks the tetrahedral carbon and another chloride ion departs, leading to interconversion between states **1 ↔ 2**. We set two C–Cl bond distances as CVs and apply restraints to prevent chloride ions from diffusing away, ensuring the CV space remains bounded. Given the small system size, a single configuration (state **1**) suffices for pretraining the sampler. Using WT-ASBS, we obtain the 2-D PMF along the two bond distances (Fig. 4b). The symmetric PMF agrees well with the WTMetaD result (Fig. 16), and both the location and free energy barrier of the transition state (TS) coincide with the results from standard saddle-point optimization.

**Post-transition-state bifurcation** For simple reactions like $S_N2$ with a single reaction channel between reactants and products, the process can be well characterized by identifying the TS and evaluating single-point energies, with harmonic approximations describing the local energy landscape around the reaction progress (Truhlar et al., 1996; Peters, 2017). However, many realistic challenges in chemistry involve branched or multimodal energy landscapes that cannot be captured by such approximations and instead requires explicit sampling. The reaction in Fig. 4c exemplifies

this difficulty: reactant **3** passes through a single TS region that bifurcates into two products **4** and **5**. Because the TS is not uniquely associated with a single product (i.e., the TS is *ambimodal*), characterizing this reaction requires sampling the full reactive landscape.

To construct CVs, we represent the three relevant C–C distances (Fig. 4d) as normalized contacts in Eq. (55), capturing bond formation progress (Fig. 4e). From these we define two CVs: $s_1 = c_1 + c_2 + c_3$, which tracks overall bond formation (a *soft* bond count), and $s_2 = c_2 - c_3$, which distinguishes between products **4** and **5**. Details on restraints and settings are provided in the Section D. Fig. 4f and g show the PMFs $F(s_1)$ and $F(s_1, s_2)$, and similarly to $S_N2$ example, the PMFs agree well with the WTMetaD result (2-D comparison in Fig. 17), providing accurate energetics for complex, multimodal reactive landscapes that dominate practical challenges in chemistry. Further discussion on reaction sampling tasks is provided in Section F.

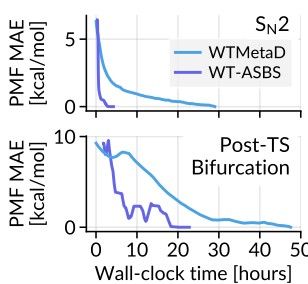

**Computational efficiency** We compared wall-clock time on four A100 80GB GPUs, where the MAE of the current bias relative to the final one, defined in Eq. (57), is shown in Fig. 5. For the $S_N2$ reaction, WT-ASBS required 0.77M energy evaluations and 4.3 hours, versus 4.0M and 29 hours for WTMetaD. For the bifurcation reaction, WT-ASBS used 2.6M evaluations and 23 hours, compared to 6.4M and 48 hours. Although WT-ASBS integrates SDEs with a control model, it needs fewer energy evaluations, which can be batched during training, effectively *distilling* a lighter sampler model from the heavier uMLIP.

Figure 5: **Convergence of PMFs.** PMF MAE from the final converged PMF for each reaction and method, compared over wall-clock time.

## 5 DISCUSSION

We presented WT-ASBS, which integrates well-tempered bias deposition over chosen CVs into a diffusion-based sampler, enabling exploration of unseen conformational modes and recovering PMFs and importance sampling weights that reconstruct the Boltzmann distribution. By embedding enhanced sampling directly into the sampler, WT-ASBS links diffusion-based generative samplers with classical rare event techniques and provides a practical recipe for sampling molecular systems with controllable bias strength and straightforward statistical reweighting. We have also demonstrated that WT-ASBS can be combined with ML CVs (Section F.4), illustrating that the method is compatible with data-driven representations and can be used as a downstream sampler once informative low-dimensional coordinates have been learned. Looking ahead, we expect the most effective workflows to combine energy-based sampling with data-based generative models and to mix global moves (diffusion-based proposals in configuration or CV space) with local sampling (short MD or MCMC) in a problem-dependent way, for example, using global diffusion moves for slow solute rearrangements and local samplers for fast solvent motions. Finally, although this work focuses on sampling molecular configurations, low-dimensional latent variables have proven useful in discrete lattice sampling (Swendsen & Wang, 1987; Wang & Landau, 2001) and dynamic Bayesian networks (Doucet et al., 2000), suggesting that extending the WT-ASBS-like approach beyond molecular systems would be a fruitful direction.

**Limitations** WT-ASBS requires the specification of CVs, and we primarily utilized predefined CVs and extended to ML CVs where metastable states are known. More complex systems will benefit from ML CVs and iterative cycles that interleave exploratory WT-ASBS runs with CV training updates. A fair comparison of enhanced sampling methods remains challenging due to the large number of algorithmic and system-specific hyperparameters (Hénin et al., 2022), and although WTMetaD may not be the optimal baseline for every system, we use it here as a natural baseline because our bias deposition scheme is inherited from WTMetaD. Future work will explore combinations of different enhanced sampling strategies to provide a more comprehensive analysis. Finally, WT-ASBS does not provide amortized sampling across systems or thermodynamic conditions, and extending it with a variationally parameterized bias (Bonati et al., 2019) or shared control networks across related systems is a promising direction for transferring biases between systems that share similar slow modes.

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

# Appendix

## A  Additional Background

### A.1  Enhanced Sampling

Efficiently sampling molecular configurations requires overcoming high free energy barriers that cause rare but important states to be visited infrequently. Enhanced sampling methods are a class of techniques developed in the computational chemistry community to address this challenge. The core idea is to bias the simulation dynamics to encourage exploration of less-populated regions of phase space, while applying a compensating weight to recover correct thermodynamic averages. Unlike unbiased MD or MCMC (which can get trapped in deep energy wells), enhanced sampling methods add external forces or modify the energy surface so that the system escapes metastable states more readily.

One foundational approach is umbrella sampling. Introduced by Torrie & Valleau (1977), umbrella sampling runs simulations with an added bias potential $U(s)$ along a chosen reaction coordinate or collective variable $s = \xi(x)$. By choosing a series of overlapping bias "umbrellas" that cover the range of $\xi$, one can force the system to sample even high-energy configurations in each window.

Each window's samples are then reweighted by $\exp(+\beta U(s))$ to reconstruct the unbiased free energy landscape. This method demonstrated the principle of biasing along important coordinates to access otherwise inaccessible states. However, it requires a careful tuning of windows.

Metadynamics took adaptive biasing a step further by dynamically building the bias potential during a single simulation. In the original metadynamics (Laio & Parrinello, 2002), the simulation deposits small Gaussian hills on the free energy landscape along the chosen CVs every few timesteps. These hills accumulate in wells (low free energy regions), gradually filling them and effectively raising the energy floor so the system escapes. Over time, the bias $V_t(s)$ approximates the negative of the underlying free energy $-F(s)$, flattening the CV landscape. Well-tempered metadynamics (Barducci et al., 2008) refined this approach by gradually reducing the height of added hills as the bias grows. This means early in the simulation, the bias grows quickly (promoting exploration), but later it converges towards a finite bias that yields a modified stationary distribution rather than diverging. Specifically, well-tempering introduces a bias factor $\gamma > 1$ so that the deposited bias is scaled by a factor that depends on the current bias (preventing overfill). The resulting steady-state is the well-tempered distribution in CV space, in which the free energy barriers are lowered by a factor of $\gamma$, effectively simulating the system at a higher temperature $T_{\text{eff}} = \gamma T$ along those CVs. Importantly, even with bias, samples can be reweighted to recover the true Boltzmann distribution. Well-tempered metadynamics thus balances exploration and accuracy by never completely flattening the landscape, avoiding infinitely large weights and ensuring the bias converges asymptotically to a finite correction.

In the enhanced sampling community, there have been continual improvements to biasing algorithms. Notably, on-the-fly probability enhanced sampling (OPES) by Invernizzi & Parrinello (2020) is a performant metadynamics variant where the bias is constructed by directly targeting a predefined distribution (often the well-tempered distribution) at each step rather than depositing fixed-shape hills. This results in smoother bias updates and often faster convergence. Although we do not explicitly implement OPES, our algorithm likewise targets the well-tempered distribution Eq. (7). In a sense, WT-ASBS could incorporate OPES-style updates (adjusting the bias based on deviation between current CV histogram and target distribution), which could be an interesting direction for future work. Additionally, techniques like variationally optimized enhanced sampling (VES) (Valsson & Parrinello, 2014) cast bias finding as an optimization problem of $V(s)$ to minimize the KL divergence between biased and unbiased ensembles. Such approaches could also be combined with our generative sampler by using the model's output to estimate that KL objective. In our current work, we use the simpler hill-based scheme inspired by WTMetaD.

Beyond metadynamics, numerous enhanced sampling strategies exist. The adaptive biasing force (ABF) method (Darve et al., 2008) takes a different approach: instead of depositing hills, it continuously estimates the mean force along the CV and subtracts it, effectively flattening the free energy gradient. ABF and its extensions (e.g., extended ABF in Comer et al. (2015)) guarantee convergence by driving the observed force toward zero in well-sampled regions. In practice, ABF and metadynamics often achieve similar outcomes: a nearly uniform sampling of $s$, but via force-balancing vs. potential-filling mechanisms. Another family of methods is replica exchange and parallel tempering. In replica exchange MD (Sugita & Okamoto, 1999), multiple copies of the system are simulated at different temperatures (or Hamiltonians) in parallel. These replicas occasionally swap configurations, allowing a cold replica to receive a configuration from a hot replica that has overcome barriers. Over many swaps, each replica samples from its designated ensemble (e.g., the lowest-temperature one samples the target distribution) but benefits from exchanges to traverse phase space faster. Parallel tempering effectively raises the chance of barrier crossing by leveraging high-temperature dynamics, albeit at the computational cost of running many simulations. It is a powerful method to explore complex landscapes, though it requires careful selection of the temperature ladder and has high computational overhead if many replicas are needed.

## A.2 NEURAL SAMPLERS

**Historical developments**   As opposed to MCMC and MD methods, generative models can potentially be used to generate samples from multimodal distributions without getting trapped in local minima or requiring lengthy equilibration times. Using generative models to sample probability distributions specified by a potential function was originally proposed by Noé et al. (2019); Albergo et al. (2019), and Nicoli et al. (2020) in the contexts of many-body physics, lattice field theory, and

statistical mechanics, respectively. These initial approaches trained discrete-time normalizing flows (Tabak & Turner, 2013) by optimizing the KL divergence between the model and the unnormalized target densities. As the use of reverse-KL objective is prone to mode collapse, the community explored alternative training objectives for flow-based sampling (Vaitl et al., 2022; Midgley et al., 2022; Máté & Fleuret, 2023). However, using flow-based methods with these approaches presents challenges. Computing the likelihood of continuous flow models (Chen et al., 2018) is computationally expensive, yet it is a necessary component of many of these potential function-based learning objectives. Meanwhile, designing expressive discrete-time normalizing flows is complex due to the restrictions on their coupling layers (Dinh et al., 2016; Midgley et al., 2023).

Thinking of the sampling task as a problem of measure transport opens up other lines of research which make use of ideas introduced by denoising diffusion models (Sohl-Dickstein et al., 2015; Ho et al., 2020) and related, more general frameworks for generative modeling (Lipman et al., 2022; Albergo et al., 2023). In data-based settings, these approaches use intermediate distributions to transport samples to the target distribution, guided by a local quadratic regression ("matching") objective that ensures the correct change of measure. The simulation-free nature of these approaches, combined with their straightforward training objectives, makes them more scalable and easier to train than previous methods, establishing them as the leading generative paradigm today. In the data-free setting, however, since samples are not available, trajectories must be sampled from the model. The central goal of recent works has thus been to design training objectives that (i) lead to a correct transport of measures and (ii) do not require backpropagation through the learned dynamics.

Vargas et al. (2023a); Berner et al. (2022) optimize the KL divergence in path space and establish connections between generative modeling and SOC. Zhang & Chen (2021); Havens et al. (2025) also approach the problem from the viewpoint of SOC by minimizing the control objective with the terminal cost defined by the target energy. Richter & Berner (2023) and Vargas et al. (2023b) optimize the variance of Radon–Nikodym derivative between the forward and backward path measures. Albergo & Vanden-Eijnden (2024) generalizes annealed importance sampling (AIS) by learning an SDE that minimizes the impact of unbiasing weights of AIS.

**Related works**   Recent methods attempt to alleviate the challenge of energy-based training struggling to capture multiple modes in complex energy landscapes. Schopmans & Friederich (2025) applied temperature annealing in Boltzmann Generators, and concurrently Akhound-Sadegh et al. (2025); Rissanen et al. (2025) incorporated inference-time annealing into diffusion models to sample from Boltzmann densities. These approaches smoothly interpolate from a high-temperature (smoothed-out) version of the target distribution to the true low-temperature distribution akin to parallel tempering. Our method biases the sampler on the fly along specific CVs, which can be seen as a more targeted form of annealing that raises the effective sampling temperature only along selected important degrees of freedom.

Another line of related work focuses on directly encouraging exploration during training. Kim et al. (2025) improved training efficiency by incorporating MCMC with bias from random network distillation during the search stage, steering exploration toward configurations that would otherwise be rarely visited, conceptually similar to the goal of amplifying exploration. In recent work, Blessing et al. (2025) formulate a diffusion-based sampler with a trust-region constrained optimal control perspective, which effectively adds Lagrange multiplier terms to enforce constraints on the path distribution. Our approach is unique in that it incorporates an adaptive bias during training that specifically targets low-dimensional CVs, which is highly effective for exploring rare-event regions and further enables reconstruction of the PMF.

### A.3   EXTENDED RELATED WORKS

**ML-based transition path sampling**   While our method is designed for equilibrium sampling with a CV-based biasing inside an energy-based diffusion sampler, closely related ideas have been developed for sampling rare dynamical trajectories by learning control forces on trajectory space, in the context of transition path sampling (TPS). Holdijk et al. (2023) cast molecular TPS as an SOC problem and learn CV-free policies that steer trajectories between metastable states. Du et al. (2024) propose Doob's Lagrangian, a variational formulation of Doob's $h$-transform (Doob, 1957) that learns conditional dynamics for generating transition paths between prescribed endpoints. Seong et al. (2025) introduce a diffusion-based TPS with off-policy training based on log-variance di-

vergence between biased MD and target path measures. Different variational approaches to path sampling of rare events are reviewed in Singh et al. (2025), with a focus on trajectory ensemble reweighting and optimal control approaches. Finally, Raja et al. (2025) repurpose pretrained generative models for TPS by interpreting paths as trajectories of an induced SDE and finding most-probable transition paths via Onsager–Machlup action minimization.

**Data-based diffusion sampling in chemistry** Complementary to our energy-based sampler that only assumes access to the potential energy and its gradients, several works train diffusion or flow-based generative models directly on molecular simulation or thermodynamic data to build data-driven equilibrium samplers. Wang et al. (2022) train diffusion models on ensembles at multiple thermodynamic conditions to generate molecular configurations across temperature or parameter space. Herron et al. (2024) develop thermodynamic maps, a diffusion-like generative approach to model the joint distribution of configurations and auxiliary thermodynamic variables to reconstruct phase behavior and other macroscopic properties from sparse data. Qiu et al. (2025) develop latent thermodynamic models that jointly learn low-dimensional representations and temperature-dependent equilibrium distributions. Benayad & Stirnemann (2025) combine replica exchange simulations with diffusion models and importance sampling to refine conformational ensembles and explore free-energy landscapes along chosen CVs.

## A.4 MOLECULAR ISOMERISM

This appendix specifies how we operationalize chemical "sameness" and "difference" in terms of minimal geometric features, their coordination numbers at the atoms that define each stereochemical element, and the order parameter restraints we apply during sampling. We adopt the IUPAC taxonomy (McNaught & Wilkinson, 1997), which separates constitutional isomerism from stereoisomerism and further divides stereoisomerism into configurational and conformational classes.

- *Constitutional isomers* share an elemental formula but differ in bond connectivity and therefore are different compounds. In our default setup, they are out of scope, except when the declared reaction of interest explicitly includes bond formation or breaking.

- *Stereoisomers* have the same connectivity but differ in three-dimensional arrangement. They are subdivided into:
  - *Configurational isomers*, which are separated by large barriers and are isolable on experimental time scales. This category includes absolute configuration at stereogenic centers (R/S), geometric isomerism such as E/Z, and helical forms. When only a single stereocenter is inverted, R and S related molecules are enantiomers; with multiple stereocenters, the possibilities include diastereomers.
  - *Conformational isomers*, which interconvert over low barriers, for example, through rotation about single bonds or ring flips.

This taxonomy is paired with barrier and half-life delineations that mark fluxional, practically isolable, and configurationally stable regimes (Canfield et al., 2018).

**Accessible component $\mathcal{A}$** Let $\mathcal{X}$ denote the configurational space for a fixed chemical composition. We select a reference configuration and restrict attention to the portion of $\mathcal{X}$ relevant to the molecular process under study, which is the sense in which the main text employs the term accessible component. One can regard configurations at temperature $T$ and study time horizon $\tau$ as grouped by an equivalence relation $\sim_\tau$: two configurations belong to the same class if they differ only by stereochemical elements that interconvert with a half-life shorter than $\tau$, while connectivity is preserved unless the chemical transformation of interest explicitly allows bond changes. The accessible component $\mathcal{A}$ is then the connected set, under the physical dynamics, in the quotient space $\mathcal{X}/\sim_\tau$ that contains the chosen reference configuration. Because the physical dynamics are stochastic, this construction relies on a separation of timescales, which is well supported by the exponential scaling of rates with activation energies and by established isomerism taxonomies.

**Isomerism, geometry, and order-parameter restraint** The polytope formalism in Canfield et al. (2018) provides a coordination-centric view of stereochemical change and shows how fundamental isomerization processes can be enumerated for each coordination number. In our context, this

motivates representing each stereochemical element by the smallest stable geometric feature that distinguishes its alternatives, together with a numerical order parameter defined on those features. The order parameter then becomes the target of a light, topology-preserving restraint. Here, we exemplify this with two main types of restraints considered in this work, bond restraint and tetrahedral center chirality restraint. While not discussed here, other types of isomerism, such as axial, planar, and helical chirality, can be handled analogously.

(1) *Bond restraint*: The preservation of molecular topology, i.e., the fixed bond graph of the system, should be encoded explicitly as a restraint whenever bond connectivity is not supposed to change. The exceptions are when the simulation potential already contains topological information (e.g., non-reactive classical force fields) or the scientific question explicitly involves bond making or bond breaking. Let $r_{ij}(X) = \|x_i - x_j\|$ be the instantaneous bond length between atoms $i$ and $j$. For each bonded pair or non-bonded pair to be preserved, we apply a bond restraint or a no-bond restraint, respectively:

$$U_{ij}^{\text{bond}}(X) = \frac{1}{2}\kappa[r_{ij}(X) - r_{\max}]_+^2, \tag{14}$$

$$U_{ij}^{\text{no-bond}}(X) = \frac{1}{2}\kappa[r_{\min} - r_{ij}(X)]_+^2, \tag{15}$$

where $[z]_+ := \max(z, 0)$. Here, we set $r_{\max} = 1.3(r_i^{\text{cov}} + r_j^{\text{cov}})$, $r_{\min} = 0.7(r_i^{\text{vdW}} + r_j^{\text{vdW}})$, and set the force constant $\kappa = 100$ eV/Å$^2$, with $r_i^{\text{cov}}$ and $r_i^{\text{vdW}}$ denoting the covalent and van der Waals radii of atom $i$, respectively. For examples in this work, we found that no-bond restraint Eq. (15) could be omitted in cases where valency of atoms is well-defined by the base potential energy (see implementation details in Section D).

(2) *Tetrahedral center chirality restraint*: We define the torsion (dihedral) angle of atoms $i$, $j$, $k$, and $l$, $\phi_{ijkl}(X) \in (-\pi, \pi]$ as follows:

$$\phi_{ijkl}(X) = \text{atan2}\left(b_2 \cdot ((b_1 \times b_2) \times (b_2 \times b_3)), |b_2|(b_1 \times b_2) \cdot (b_2 \times b_3)\right), \tag{16}$$

where the displacement vectors are defined as $b_1 := x_j - x_i$, $b_2 := x_k - x_j$, and $b_3 := x_l - x_k$. A *proper* torsion is the signed angle between two planes that share a central bond, with three bonds in sequence, e.g., $i$–$j$–$k$–$l$, with the torsion angle describing the relative rotation around the bond $j$–$k$. In contrast, an *improper* torsion does not describe rotation around a bond, but rather the out-of-plane displacement of one atom relative to a reference plane defined by three others.

We now consider a tetrahedral stereocenter $i$ with three neighboring atoms $j$, $k$, and $l$ (out of its four substituents). The corresponding improper torsion angle, $\phi_{ijkl}(X)$, exhibits a bimodal distribution: each mode corresponds to a distinct stereoisomer, and in the ideal tetrahedral geometry, the peaks are centered near $\pm 35°$. To preserve the stereochemical configuration, we introduce the following flat-bottom harmonic restraint potential on the improper torsion:

$$U_{ijkl}^{\text{imp}}(X; \kappa, \phi_0, \phi_{\text{tol}}) = \frac{1}{2}\kappa[|w(\phi_{ijkl}(X) - \phi_0)| - \phi_{\text{tol}}]_+^2 \tag{17}$$

where $w(\alpha) = \text{atan2}(\sin\alpha, \cos\alpha) \in (-\pi, \pi]$ is the $2\pi$-periodic wrapping function, $\phi_0$ is the reference torsion value, and $\phi_{\text{tol}}$ is the flat-bottom tolerance.

For clarity, we illustrate this construction using the $C_\alpha$ stereocenter of an amino acid. In naturally occurring L-amino acids, the improper torsion defined by $(C_\alpha, N, C, C_\beta)$ is positive, whereas the improper torsion $(C_\alpha, N, C, H_\alpha)$ is negative. Based on this, we define the amino acid chirality restraint as

$$U_{\text{AA}}^{\text{chiral}}(X) = U_{C_\alpha, N, C, C_\beta}^{\text{imp}}(X; \kappa, +\phi_0, \phi_{\text{tol}}) + U_{C_\alpha, N, C, H_\alpha}^{\text{imp}}(X; \kappa, -\phi_0, \phi_{\text{tol}}), \tag{18}$$

where we set $\kappa = 25.0$ eV/rad$^2$, $\phi_0 = 0.615$ rad ($35°$), and $\phi_{\text{tol}} = 0.436$ rad ($25°$).

# B  PROOFS AND NOTES

## B.1  PROOF FOR PROPOSITION 3.1

**Proposition 3.1** (Convergence of WT-ASBS). *When Algorithm 1 is performed until convergence, the bias potential $V_k$ converges almost surely to $V^*(s) = -(1 - \frac{1}{\gamma})F(s) + \text{const}$, and hence the sampled distribution converges to the well-tempered target Eq. (7).*

*Proof.* We follow Dama et al. (2014) for the bias decomposition and stochastic approximation framework, extending it with WT-ASBS-specific derivations and clearer notation.

*Step 1: Level and driving parts of the bias and the step size.* We split the bias $V_k(s)$ into a *level* $L_k := \frac{1}{|\mathcal{S}|} \int_{\mathcal{S}} V_k(s) \, ds$ and a *driving part* $D_k(s) := V_k(s) - L_k$. Only $D_k$ affects the biased distribution, and adding the scalar $L_k$ shifts the energy by a constant. Now, a single step of well-tempered bias update can be split into

$$L_{k+1} = L_k + \exp\left(-\frac{\beta}{\gamma - 1} L_k\right) \Lambda[D_k](s_{k+1}), \tag{19}$$

$$D_{k+1}(s) = D_k(s) + \exp\left(-\frac{\beta}{\gamma - 1} L_k\right) \Gamma[D_k](s, s_{k+1}), \tag{20}$$

where we define $\Lambda[D](s')$ and $\Gamma[D](s, s')$ as follows:

$$\Lambda[D](s') := \frac{1}{|\mathcal{S}|} \int_{\mathcal{S}} h K_\sigma(s, s') \exp\left(-\frac{\beta}{\gamma - 1} D(s')\right) \, ds, \tag{21}$$

$$\Gamma[D](s, s') := h K_\sigma(s, s') \exp\left(-\frac{\beta}{\gamma - 1} D(s')\right) - \Lambda[D](s'). \tag{22}$$

Now, under mild practical conditions, the driving bias $D_k(s)$ stays bounded, and $\exp(-\frac{\beta}{\gamma-1} D_k(s'))$ is bounded above and below by positive constants. Also, $\int_{\mathcal{S}} h K_\sigma(s, s') \, ds$ is a continuous positive function of $s'$ on a compact set, so it also has positive finite bounds. Therefore, from Eq. (21), we get the following bound with positive constant $a$ and $b$:

$$a \le \Lambda[D_k](s_{k+1}) \overset{(19)}{=} (L_{k+1} - L_k) \exp\left(\frac{\beta}{\gamma - 1} L_k\right) \le b. \tag{23}$$

This result implies that the asymptotic effective step size is bounded by positive multiples of $\frac{1}{k}$: $\epsilon_k := \exp(-\frac{\beta}{\gamma-1} L_k) \asymp \frac{1}{k}$.

*Step 2: Stochastic recursion to a mean-field ODE.* Now, we define an auxiliary time scale $\tau$ with $\tau_0 = 0$ and $\tau_{k+1} = \tau_k + \epsilon_k$. Then, the driving part update Eq. (20) can be written in stochastic approximation form

$$D_{k+1}(s) = D_k(s) + \epsilon_k(\Gamma[D_k](s) + \zeta_{k+1}(s)), \tag{24}$$

where we rewrite Eqs. (21) and (22) with expectation over $S \sim \bar{\nu}_D$, the CV marginal under the bias $V = L + D$,

$$\Lambda[D] := \frac{1}{|\mathcal{S}|} \int_{\mathcal{S}} \mathbb{E}_{S \sim \bar{\nu}_D}\left[h K_\sigma(s, S) \exp\left(-\frac{\beta}{\gamma - 1} D(S)\right)\right] \, ds, \tag{25}$$

$$\Gamma[D](s) := \mathbb{E}_{S \sim \bar{\nu}_D}\left[h K_\sigma(s, S) \exp\left(-\frac{\beta}{\gamma - 1} D(S)\right)\right] - \Lambda[D], \tag{26}$$

and $\zeta_{k+1}$ is a martingale difference with bounded second moment. The mini-batch hill update in Algorithm 1 provides an unbiased estimator of the expectation $\mathbb{E}_{S \sim \bar{\nu}_D}$. From Step 1, $\epsilon_k \asymp \frac{1}{k}$, which satisfies the Robbins–Monro step condition $\sum_k \epsilon_k = \infty$ and $\sum_k \epsilon_k^2 < \infty$ (Robbins & Monro, 1951). Standard stochastic approximation arguments then imply that the polygonal interpolation $\hat{D}(\tau_k) = D_k$, $\hat{D}(\tau) = D_k + \frac{\tau - \tau_k}{\epsilon_k}(D_{k+1} - D_k)$ for $\tau \in [\tau_k, \tau_{k+1})$ is an asymptotic pseudo-trajectory of the mean-field ODE $\frac{d}{d\tau} D(\tau) = \Gamma[D(\tau)]$ (Kushner & Yin, 2003). Physically, this corresponds to the long-time limit equation in Dama et al. (2014), obtained by aggregating many small hill updates over a fixed $\tau$ window while the bias changes only slightly within that window.

*Step 3: Fixed points of the asymptotic ODE.* At a fixed point $D^*$, we have $\Gamma[D^*] \equiv 0$. From Eq. (26), this correspond to

$$\int_{\mathcal{S}} h K_\sigma(s, s') \exp\left(-\frac{\beta}{\gamma - 1} D^*(s')\right) \bar{\nu}_{D^*}(s') \, \mathrm{d}s' = \text{const.} \quad \forall s \in \mathcal{S}. \tag{27}$$

For Gaussian $K_\sigma$ on a compact domain, the associated integral operator is positive definite. Hence, the solution is unique and satisfies

$$\exp\left(-\frac{\beta}{\gamma - 1} D^*(s)\right) \underbrace{\exp(-\beta(F(s) + D^*(s)))}_{\propto \bar{\nu}_{D^*}(s)} = \text{const.} \tag{28}$$

$$\implies \quad V^*(s) = L^* + D^*(s) = -\left(1 - \frac{1}{\gamma}\right) F(s) + \text{const.} \tag{29}$$

*Step 4. Global stability.* Now, define the tempering-reweighted instantaneous CV density

$$p_w(s; \tau) \propto \exp\left(-\frac{\beta}{\gamma - 1} D(s; \tau)\right) \underbrace{\exp(-\beta(F(s) + D(s; \tau)))}_{\propto \bar{\nu}_D(s)}. \tag{30}$$

Let $p_w^*$ denote the target tempering-reweighted distribution, i.e., a normalized solution of

$$\int_{\mathcal{S}} h K_\sigma(s, s') p_w^*(s') \, \mathrm{d}s' = \text{const.} \tag{31}$$

For Gaussian hills on a compact $\mathcal{S}$, this solution is unique and equals the uniform density $p_w^*(s) = |\mathcal{S}|^{-1}$. Consider the Lyapunov functional defined as a KL divergence between $p_w^*$ and $p_w(\cdot; \tau)$:

$$\mathcal{D}(\tau) := D_{\mathrm{KL}}(p_w^* \| p_w(\cdot; \tau)) = \int_S p_w^*(s) \log \frac{p_w(s; \tau)}{p_w^*(s)} \, \mathrm{d}s. \tag{32}$$

We now show that $\frac{\mathrm{d}}{\mathrm{d}\tau} \mathcal{D}(\tau) \leq 0$. First,

$$\frac{\mathrm{d}}{\mathrm{d}\tau} D(s; \tau) = \Gamma[D(\tau)](s) \tag{33}$$

$$\overset{(26)}{=} \int_{\mathcal{S}} h K_\sigma(s, s') \exp\left(-\frac{\beta}{\gamma - 1} D(s'; \tau)\right) \bar{\nu}_{D(\tau)}(s') \, \mathrm{d}s' - \Lambda[D(\tau)] \tag{34}$$

$$\overset{(30)}{=} c(\tau) \int_{\mathcal{S}} h K_\sigma(s, s') p_w(s'; D_k) \, \mathrm{d}s' - \Lambda[D(\tau)] \tag{35}$$

$$\overset{(31)}{=} c(\tau) \int_{\mathcal{S}} h K_\sigma(s, s') [p_w(s'; \tau) - p_w^*(s')] \, \mathrm{d}s' - C(\tau), \tag{36}$$

where $c(\tau) > 0$ and $C(\tau)$ are constants in $s$. Also, from Eq. (30), we get

$$\frac{1}{p_w(s; \tau)} \frac{\mathrm{d}}{\mathrm{d}\tau} p_w(s; \tau) = -\frac{\beta\gamma}{\gamma - 1} \left(\frac{\mathrm{d}}{\mathrm{d}\tau} D(s; \tau) - \mathbb{E}_{p_w}\left[\frac{\mathrm{d}}{\mathrm{d}\tau} D(\cdot; \tau)\right]\right). \tag{37}$$

Finally, plugging Eqs. (36) and (37) into the time derivative of Eq. (32) yields

$$\frac{\mathrm{d}}{\mathrm{d}\tau} \mathcal{D}(\tau) \overset{(32)}{=} -\int_{\mathcal{S}} p_w^*(s) \frac{1}{p_w(s; \tau)} \frac{\mathrm{d}}{\mathrm{d}\tau} p_w(s; \tau) \, \mathrm{d}s \tag{38}$$

$$\overset{(37)}{=} \frac{\beta\gamma}{\gamma - 1} \left(\mathbb{E}_{p_w^*}\left[\frac{\mathrm{d}}{\mathrm{d}\tau} D(\cdot; \tau)\right] - \mathbb{E}_{p_w}\left[\frac{\mathrm{d}}{\mathrm{d}\tau} D(\cdot; \tau)\right]\right) \tag{39}$$

$$\overset{(36)}{=} -\frac{\beta\gamma}{\gamma - 1} c(\tau) \int_{\mathcal{S}} \int_{\mathcal{S}} [p_w(s; \tau) - p_w^*(s)] h K_\sigma(s, s') [p_w(s'; \tau) - p_w^*(s')] \, \mathrm{d}s \mathrm{d}s' \leq 0, \tag{40}$$

since the integral operator with the Gaussian kernel $h K_\sigma(\cdot, \cdot)$ is positive semidefinite and $c(\tau) > 0$. The equality in Eq. (40) holds if and only if $p_w(\cdot; \tau) = p_w^*$, and $\mathcal{D}$ is a strict Lyapunov functional and the ODE solution converges to the unique fixed point.

Hence, the mean-field ODE $\frac{\mathrm{d}}{\mathrm{d}\tau} D(\tau) = \Gamma[D(\tau)]$ has a globally attracting equilibrium $D^*$, and one gets $D_k \to D^*$ and $V_k \to V^*$ in Eq. (29) almost surely by following the stochastic approximation Eq. (24). □

# C  IMPLEMENTATION DETAILS FOR WT-ASBS

## C.1  WT-ASBS WITH REPLAY BUFFER

Here, we describe the practical implementation of Algorithm 1, which merges the well-tempered bias update into the AM replay buffer update within the inner step. The resulting WT-ASBS algorithm with replay buffer is shown in Algorithm 2. After updating the AM replay buffer $\mathcal{B}_{\text{adj}}$ (line 6), we reuse the same samples, projecting them to CV space to obtain the CV values for Gaussian kernel deposition (line 7).

---

**Algorithm 2** WT-ASBS with replay buffer

---

**Require:** prior $\mu$, potential energy $E(x)$, restraint $U(x)$, CV $\xi(x)$, control $u_\theta(x,t)$, corrector $h_\phi(x)$, replay
buffers $\mathcal{B}_{\text{adj}}$ and $\mathcal{B}_{\text{crt}}$

1: Initialize $V(s) \leftarrow 0, h_\phi^{(0)} \leftarrow 0$             ▷ *Bias and IPF initialization*

2: **for** Stage $k = 1, 2, \cdots, K$ **do**

3:      **for** AM epoch $m = 1, \cdots, M_{\text{adj}}$ **do**            ▷ *Adjoint Matching with WT bias update*

4:          Sample from model $\{(X_0^{(i)}, X_1^{(i)})\}_{i=1}^N \sim p^{\bar{u}^{(k)}}$, where $\bar{u}^{(k)} = \texttt{stopgrad}(u_\theta^{(k)})$

5:          Compute adjoint target $a_t^{(i)} \leftarrow \texttt{stopgrad}(\texttt{clip}(\nabla(E + U + V \circ \xi)(X_1^{(i)}), \alpha_{\max}) + h_\phi^{(k)}(X_1^{(i)}))$

6:          Update replay buffer $\mathcal{B}_{\text{adj}} \leftarrow \mathcal{B}_{\text{adj}} \cup \{(X_0^{(i)}, X_1^{(i)}, a_t^{(i)})\}_{i=1}^N$

7:          Update bias with sampled CVs $s^{(i)} \leftarrow \xi(X_1^{(i)})$:

$$V(s) \leftarrow V(s) + h \sum_{i=1}^N \exp(-\tfrac{\beta}{\gamma-1} V(s^{(i)})) \exp(-\tfrac{\|s-s^{(i)}\|^2}{2\sigma^2}) \tag{41}$$

8:          Take $L$ gradient steps $\nabla_\theta \mathcal{L}_{\text{AM}}$ w.r.t. the AM objective:

$$\mathcal{L}_{\text{AM}}(\theta) \leftarrow \mathbb{E}_{t \sim \mathcal{U}[0,1], (X_0, X_1, a_t) \sim \mathcal{B}_{\text{adj}}, X_t \sim p^{\text{base}}(\cdot | X_0, X_1)} \left[ \lambda_t \| u_\theta^{(k)}(t, X_t) + \sigma_t a_t \|^2 \right] \tag{42}$$

9:      **for** CM epoch $m = 1, 2, \cdots, M_{\text{crt}}$ **do**             ▷ *Corrector Matching*

10:         Sample from model $\{(X_0^{(i)}, X_1^{(i)})\}_{i=1}^N \sim p^{\bar{u}^{(k)}}$, where $\bar{u}^{(k)} = \texttt{stopgrad}(u_\theta^{(k)})$

11:         Update replay buffer $\mathcal{B}_{\text{crt}} \leftarrow \mathcal{B}_{\text{crt}} \cup \{(X_0^{(i)}, X_1^{(i)})\}_{i=1}^N$

12:         Take $L$ gradient steps $\nabla_\phi \mathcal{L}_{\text{CM}}$ w.r.t. the CM objective:

$$\mathcal{L}_{\text{CM}}(\phi) \leftarrow \mathbb{E}_{(X_0, X_1) \sim \mathcal{B}_{\text{crt}}} \left[ \| h_\phi^{(k)}(X_1) - \nabla_{X_1} \log p^{\text{base}}(X_1 | X_0) \|^2 \right] \tag{43}$$

13: **output** Final control $u_\theta^*(x, t)$, final bias $V^*(s)$

---

**Grid discretization of bias potential**    Since the bias potential is accumulated over the entire training history, longer training would require summing over an increasingly large number of Gaussian kernels to evaluate $\nabla(V \circ \xi)$. To ensure computational efficiency, we adopt the standard grid discretization approach commonly used in enhanced sampling simulations. The bias is stored on a discretized grid with spacing chosen to be smaller than the kernel width $\sigma$, and both energies and forces are obtained through interpolation between grid points.

**Pretraining for WT-ASBS**    As proposed in ASBS (Liu et al., 2025), we warm-start the training process with the approximate bridge matching objective, which aligns the control model $u_\theta$ more closely with available data samples. In our setting, these samples are generated by short MD simulations that perform local sampling around a reference configuration $x_{\text{ref}}$. The bridge matching objective $\mathcal{L}_{\text{pretrain}}$ is defined as follows:

$$\mathcal{L}_{\text{pretrain}}(\theta) = \mathbb{E}_{t \sim \mathcal{U}[0,1], X_0 \sim \mu, X_1 \sim q, X_t \sim p^{\text{base}}(\cdot | X_0, X_1)} \left[ \hat{\lambda}_t \| u_t(X_t) - \sigma_t \nabla_{X_t} \log p^{\text{base}}(X_1 | X_t) \|^2 \right], \tag{44}$$

where $q$ denotes the empirical distribution of the MD samples, and $\hat{\lambda}_t$ is a time-dependent weighting factor, chosen as $\kappa_{1|t}/\sigma_t^2$, where $\kappa_{1|t}$ is the conditional variance of the base process (see Section C.3). This pretraining is only approximate, since the marginal distribution of the optimal $u_\theta^*$ matches $q(X_1)$ exactly only when the base process is memoryless (i.e., when $X_0$ and $X_1$ are independent). Nevertheless, for warm-start purposes, this approximation is sufficient to rapidly align the model with a distribution concentrated in a single conformational mode.

## C.2 MODEL ARCHITECTURE

We build on PaiNN (Schütt et al., 2021), an E(3)-equivariant graph neural network designed for molecular energy and force prediction, with minimal modifications: adding time-conditioning inputs, incorporating vector product layers (Schreiner et al., 2023) to break parity symmetry, restricting the model to SE(3)-equivariance, and applying equivariant layer normalization (Liao et al., 2024) to improve training stability. Each atom is assigned a unique atomic embedding, as the reference configuration used to define the accessible component (Requirement 1) already specifies atom indices. The model outputs a single vector per atom, which is scaled by $\sigma_t$ to produce the final control and corrector outputs.

## C.3 BASE PROCESS

We employ the base distribution in Karras et al. (2022) (EDM), which defines $\sigma_t$ through a power-schedule parameter $\rho$ and yields an analytically integrable variance between two different times $\kappa_{t|s}$:

$$\sigma_t := \left[(1-t)\sigma_{\max}^{1/\rho} + t\sigma_{\min}^{1/\rho}\right]^\rho, \tag{45}$$

$$\kappa_{t|s} := \int_s^t \sigma_\tau^2 \, d\tau = \frac{1}{2\rho+1} \frac{\sigma_s^{2+1/\rho} - \sigma_t^{2+1/\rho}}{\sigma_{\max}^{1/\rho} - \sigma_{\min}^{1/\rho}}. \tag{46}$$

With the reparametrized time $\gamma_t := \frac{\kappa_{t|0}}{\kappa_{1|0}} \in [0,1]$, conditional base distributions for the driftless base $dX_t = \sigma_t dW_t$ are

$$p_{t|0}^{\text{base}}(X_t|X_0) = \mathcal{N}(X_t; X_0, \kappa_{t|0}I), \tag{47}$$

$$p_{t|0,1}^{\text{base}}(X_t|X_0, X_1) = \mathcal{N}(X_t; (1-\gamma_t)X_0 + \gamma_t X_1, \gamma_t(1-\gamma_t)\kappa_{1|0}I). \tag{48}$$

**Translational invariance** As in AS (Havens et al., 2025), we enforce translational invariance by projecting the SDE onto the subspace $\mathcal{X}^{\text{CoM}} = \{x \mid \sum_{i=1}^n x^i = 0\} \subset \mathcal{X}$, where $x^i \in \mathbb{R}^3$ is the position of $i$-th atom. This is achieved through the projection operator $P$, which acts on any $y \in \mathbb{R}^{n \times 3}$ by subtracting the center of mass from each coordinate $y^i$ for $i \in [n]$.

$$x = Py, \qquad P = \left(I_n - \frac{1}{n}\mathbf{1}_n\mathbf{1}_n^\top\right) \otimes I_d \tag{49}$$

$$dX_t = \sigma_t P u_t(X_t) \, dt + \sigma_t P \, dW_t, \qquad X_0 = 0 \tag{50}$$

For the base process, the dynamics are given by

$$X_t = X_0 + \int_0^t \sigma_s P \, dW_s \tag{51}$$

$$\text{Var}[X_t|X_0] = \int_0^t \sigma_s^2 PP^\top \, ds = \kappa_{t|0} PP^\top. \tag{52}$$

Sampling can be carried out by first drawing $Y_t \sim \mathcal{N}(X_0, \kappa_{t|0}I)$ and then projecting to obtain $X_t = PY_t$, which enforces the translational invariance constraint.

Adjoint matching Eq. (10) and corrector matching Eq. (11) are modified accordingly as follows:

$$u := \arg\min_u \mathbb{E}_{p_{t|0,1}^{\text{base}} p_{0,1}^{\bar{u}}}[\|P(u_t(X_t) + \sigma_t(\nabla E + h^{(k-1)})(X_1))\|^2], \; \bar{u} = \texttt{stopgrad}(u), \tag{53}$$

$$h := \arg\min_h \mathbb{E}_{p_{0,1}^{u^{(k)}}}[\|P(h(X_1) - \nabla_{X_1} \log p_{1|0}^{\text{base}}(X_1|X_0))\|^2]. \tag{54}$$

## C.4 HYPERPARAMETERS

Model and training hyperparameters for each experiment are listed in Table 1. For WT-ASBS Algorithm 2, we use $L = 100$ gradient updates per AM and CM epoch, and $\alpha_{\max} = 100.0$ to clip the energy gradient in the terminal cost. Models were trained with AdamW optimizer (Loshchilov

Table 1: **Hyperparameters for WT-ASBS.** Model and training hyperparameters for each experiment.

| Hyperparameter | Alanine dipeptide | Alanine tetrapeptide | Nucleophilic substitution | Post-TS bifurcation |
|---|---|---|---|---|
| *Models* | | | | |
| Hidden dimension | 128 | 128 | 64 | 128 |
| Radial basis | 64 | 64 | 32 | 64 |
| Control layers | 4 | 4 | 3 | 4 |
| Corrector layers | 3 | 3 | 3 | 3 |
| Radial cutoff [Å] | 8.0 | 8.0 | 8.0 | 8.0 |
| *Batch and replay buffer* | | | | |
| AM/CM buffer size | 16,384 | 4,096 | 4,096 | 4,096 |
| Buffer samples per epoch | 2,048 | 512 | 512 | 512 |
| Batch size | 2,048 | 512 | 512 | 512 |
| *Base SDE* | | | | |
| Source distribution | $\mathcal{N}(0, 1\,\text{Å}^2)$ | $\mathcal{N}(0, 1\,\text{Å}^2)$ | $\mathcal{N}(0, 1\,\text{Å}^2)$ | $\mathcal{N}(0, 1\,\text{Å}^2)$ |
| $\sigma_{\min}$ [Å] | 0.001 | 0.001 | 0.001 | 0.001 |
| $\sigma_{\max}$ [Å] | 6.0 | 6.0 | 5.0 | 6.0 |
| EDM exponent $\rho$ | 3.0 | 3.0 | 3.0 | 3.0 |
| *Bias update* | | | | |
| Initial Gaussian $h$ [eV] | $5 \times 10^{-4}$ | $5 \times 10^{-4}$ | $5 \times 10^{-4}$ | $2 \times 10^{-4}$ |
| Gaussian width $\sigma$ | [0.2, 0.2] rad | [0.3, 0.3, 0.3] rad | [0.2, 0.2] Å | [0.05, 0.05] |
| CV grid minimum | $[-\pi, -\pi]$ rad | $[-\pi, -\pi, -\pi]$ rad | [1.0, 1.0] Å | [0.0, -1.0] |
| CV grid maximum | $[+\pi, +\pi]$ rad | $[+\pi, +\pi, +\pi]$ rad | [4.0, 4.0] Å | [3.0, 1.0] |
| CV grid bins | [150, 150] | [100, 100, 100] | [150, 150] | [300, 200] |
| Periodic CV grid | True | True | False | False |
| Bias factor $\gamma$ | 8.0 | 12.0 | 12.0 | 30.0 |
| *Training* | | | | |
| Pretraining learning rate | $3 \times 10^{-4}$ | $3 \times 10^{-4}$ | $3 \times 10^{-4}$ | $3 \times 10^{-4}$ |
| Pretraining steps | 100,000 | 100,000 | 20,000 | 100,000 |
| Training stages $K$ | 4 | 40 | 2 | 4 |
| AM epochs $M_{\text{adj}}$ | 1,000 | 1,000 | 500 | 1,000 |
| CM epochs $M_{\text{crt}}$ | 500 | 500 | 250 | 500 |
| Initial learning rate | $1 \times 10^{-4}$ | $1 \times 10^{-4}$ | $3 \times 10^{-5}$ | $3 \times 10^{-5}$ |
| Final learning rate | $3 \times 10^{-5}$ | $5 \times 10^{-5}$ | $1 \times 10^{-5}$ | $1 \times 10^{-5}$ |
| Gradient clipping value | 100.0 | 100.0 | 100.0 | 100.0 |

& Hutter, 2017) using a weight decay of $10^{-3}$, and diffusion process was integrated with Euler–Maruyama method using 250 steps. For the first AM stage, we applied a 50-epoch learning rate warmup, after which bias deposition began. After completing $K$ training stages, we added a final AM stage without further bias deposition. Learning rates followed a cosine annealing schedule during training, applied independently to AM and CM optimization.

Several hyperparameters affecting training duration were selected based on the convergence of Algorithm 2. The number of gradient updates $L$ was chosen to ensure that the controller parameters could track the outer-loop bias updates, as examined further in the ablation studies in Section E. The AM and CM epochs per stage ($M_{\text{adj}}$ and $M_{\text{crt}}$) were largely adopted from previous work (Liu et al., 2025), and we found them sufficient for the AM and CM losses to stabilize within each training stage. The number of alternating training stages $K$ was set according to the convergence of the bias or PMF, assessed by whether newly deposited bias became negligible or whether the estimated free energy differences along the selected CVs had stabilized.

## D    EXPERIMENT DETAILS

For small peptides, we performed classical force field simulations with OpenMM 8.3.1 (Eastman et al., 2023), while reactions were modeled using UMA-S-1.1 uMLIP (Wood et al., 2025) as implemented in the `fairchem-core` 2.3.0 package (Shuaibi et al., 2025) with MD integrators in ASE 3.25.0 (Larsen et al., 2017). All MD-based enhanced sampling (WTMetaD) simulations were carried out with PLUMED 2.9.3 (Tribello et al., 2014; The PLUMED consortium, 2019).

### D.1    ALANINE DIPEPTIDE

**Energy model**    The potential energy of the system was described by Amber ff96 force field (Kollman et al., 1997) with GBSA-OBC implicit solvation (Onufriev et al., 2004), as implemented in OpenMM (Eastman et al., 2023).

**Restraints**    We apply the chirality restraint Eq. (18) to the central alanine, $U_{\text{Ala2}}^{\text{chiral}}(X)$, during WT-ASBS training. This restraint is not used in MD-based sampling because chirality inversion does not occur under physical dynamics.

**Pretraining data**    We generated pretraining data using a 100 ps molecular dynamics simulation at 300 K with a 4 fs time step, enabled by hydrogen bond constraints and hydrogen mass repartitioning (HMR; Hopkins et al. (2015)) with hydrogen masses set to 1.5 amu. The system was equilibrated for 10 ps before sampling $10^3$ frames at 0.1 ps intervals. Simulations employed the Langevin middle integrator (Zhang et al., 2019) as implemented in OpenMM (Eastman et al., 2023), with a collision rate of 1 ps$^{-1}$. In total, pretraining data generation required $2.75 \times 10^4$ energy evaluations.

**Reference data**    We employed the test split of $10^7$ samples from Midgley et al. (2022), generated by replica exchange MD with 21 replicas spanning 300 K to 1300 K. Their dataset generation required $2.3 \times 10^{10}$ energy evaluations.

**WTMetaD simulations**    We performed WTMetaD simulations on the backbone torsion CVs $(\phi, \psi)$. The dynamics were propagated with Langevin middle integrator (Zhang et al., 2019) at 300 K with a 1 fs timestep and 1 ps$^{-1}$ collision rate. Following a 10 ps equilibration at 300 K, Gaussian biases were deposited every 500 steps with widths of 0.2 rad for both variables. The bias update employed an initial Gaussian height of 1.2 kJ/mol, a bias factor of 8, and a grid covering $[-\pi, \pi]^2$ with $150 \times 150$ bins. Each production run was 10 ns in length, and simulations were repeated four times to ensure statistical reliability, yielding $4.0 \times 10^7$ energy evaluations.

### D.2    ALANINE TETRAPEPTIDE

**Energy model**    The potential energy of the system was described by Amber ff99SB-ILDN force field (Lindorff-Larsen et al., 2010) with Amber99-OBC implicit solvation (Onufriev et al., 2004), as implemented in OpenMM (Eastman et al., 2023).

**Restraints** We apply the chirality restraints Eq. (18) to three alanine residues, $U_{\text{Ala2}}^{\text{chiral}}(X) + U_{\text{Ala3}}^{\text{chiral}}(X) + U_{\text{Ala4}}^{\text{chiral}}(X)$, during WT-ASBS training. As in the alanine dipeptide case, these restraints are not used in MD-based sampling.

**Pretraining data** We generated pretraining data using a 1 ns molecular dynamics simulation at 300 K with a 4 fs time step, enabled by hydrogen bond constraints and HMR with hydrogen masses set to 1.5 amu. The system was equilibrated for 10 ps before sampling $10^3$ frames at 1 ps intervals. Simulations employed the Langevin middle integrator (Zhang et al., 2019) as implemented in OpenMM (Eastman et al., 2023), with a collision rate of 1 ps$^{-1}$. In total, pretraining data generation required $2.52 \times 10^5$ energy evaluations.

**Reference data** We generated unbiased alanine tetrapeptide conformations by first performing parallel tempering simulations to sample initial states, followed by long unbiased molecular dynamics runs for data collection. Parallel tempering simulations were carried out with OpenMM and OpenMMTools (Chodera et al., 2025) using MPI parallelization across 8 replicas, with temperatures geometrically spaced between 300 K and 600 K. Replicas were propagated with the generalized hybrid Monte Carlo (GHMC) integrator (Stoltz et al., 2010) using a 2 fs timestep (with hydrogen bond constraints), 1 ps$^{-1}$ collision rate, and 1000 steps between exchange attempts (2 ps). Simulations were initialized from an energy-minimized structure, equilibrated for 200 ps, and then run for 100 ns of production with replica exchanges attempted every 2 ps.

From the 300 K replica, 64 independent unbiased MD simulations were seeded from randomly selected frames. Each simulation employed a Langevin middle integrator (Zhang et al., 2019) at 300 K with a 1 fs timestep and 1 ps$^{-1}$ collision rate. After 100 ps of equilibration, production simulations were performed for 1.56 µs, yielding a total of 100 µs simulation time across all replicas. Trajectories were saved every 2 ps, producing $5 \times 10^7$ samples in total. Reference data generation required approximately $1.01 \times 10^{11}$ energy evaluations.

**WTMetaD simulations** We performed WTMetaD simulations on the three backbone torsion CVs $(\phi_1, \phi_2, \phi_3)$. The same integrator settings as in the reference data generation were used. Following a 10 ps equilibration at 300 K, Gaussian biases were deposited every 500 steps with widths of 0.3 rad for each CV. The biasing scheme employed an initial Gaussian height of 1.2 kJ/mol, a bias factor of 12, and a grid covering $[-\pi, \pi]^3$ with $75^3$ bins. Each production run was 20 ns in length, and simulations were repeated four times to ensure statistical reliability, yielding $8.0 \times 10^7$ energy evaluations.

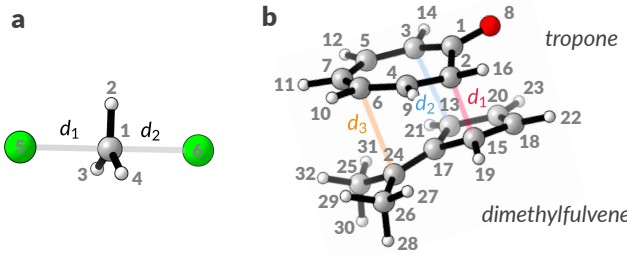

Figure 6: **Atom index definition for chemical reactions.** (a) Nucleophilic substitution (S$_\text{N}$2). (b) Cycloaddition reaction between tropone and dimethylfulvene with post-TS bifurcation.

### D.3 NUCLEOPHILIC SUBSTITUTION

**Energy model** The potential energy of the system was described by the UMA-S-1.1 (Wood et al., 2025) model with the DFT task label omol, a system charge of −1, and a singlet spin state (total spin 0).

**Restraints** In WT-ASBS, three C–H bonds (1–2, 1–3, and 1–4 in Fig. 6a) are maintained using the bond restraint in Eq. (14). For both WT-ASBS and WTMetaD, to keep the accessible CV space

$\mathcal{S}$ bounded, the same restraint form is applied to two C–Cl distances (1–5 and 1–6) with $r_{\max} = 3.3$ Å, preventing chloride ions from escaping the reaction complex.

**WTMetaD simulations**   We performed WTMetaD simulations using $(d_1, d_2)$ as the CVs. The dynamics were propagated with a Langevin thermostat at 300 K, a 0.5 fs timestep, and a friction coefficient of 10 ps$^{-1}$. Initial velocities were assigned from a Maxwell–Boltzmann distribution, and the total momentum was removed. Following a 1 ps equilibration at 300 K, Gaussian biases were deposited every 250 steps with widths of 0.2 Å for both CVs. The biasing scheme used an initial Gaussian height of 0.02 eV, a bias factor of 20, and a grid spanning $[1.0, 4.0]$ Å $\times$ $[1.0, 4.0]$ Å. Each production run was 0.5 ns in length, and simulations were repeated four times to ensure statistical reliability, yielding $4.0 \times 10^6$ energy evaluations.

### D.4   POST-TRANSITION-STATE BIFURCATION

**Energy model**   The potential energy of the system was described by the UMA-S-1.1 (Wood et al., 2025) model with the DFT task label `omol`, a system charge of 0, and a singlet spin state (total spin 0).

**Collective variables**   We define a normalized contact $c_i$ that maps the bond distance $d_i$ to a *soft* bond indicator,

$$c_i = \frac{1 - (d_i/d_{\mathrm{ref}})^6}{1 - (d_i/d_{\mathrm{ref}})^8} \in [0, 1], \tag{55}$$

which approaches 1 when $d_i$ is much smaller than $d_{\mathrm{ref}}$ and decays to 0 as $d_i$ increases (bond dissociation). To construct the CVs, we express the three relevant C–C distances (Fig. 4d) as normalized contacts using Eq. (55) with $d_{\mathrm{ref}} = 1.7$ Å, thereby capturing bond formation progress (Fig. 4e). We then define two CVs: $s_1 = c_1 + c_2 + c_3$, which measures overall bond formation, and $s_2 = c_2 - c_3$, which distinguishes between products **4** and **5**. A similar approach has been applied to enhanced sampling of mechanistically related reactions (Nam & Jung, 2023).

**Restraints**   In WT-ASBS, bonds present in the reactant configuration (**3**, Fig. 4c) are preserved using the bond restraint in Eq. (14). Furthermore, to prevent tropone (upper reactant) from "teleporting" to the opposite face of dimethylfulvene (lower reactant), we enforce an inter-planar distance defined by three atoms per plane:

$$d(X) = \underbrace{\frac{(x_1^{(1)} - x_0^{(1)}) \times (x_2^{(1)} - x_0^{(1)})}{\|(x_1^{(1)} - x_0^{(1)}) \times (x_2^{(1)} - x_0^{(1)})\|}}_{\text{Unit normal of plane 1}} \cdot \underbrace{\left( \frac{1}{3} \sum_{m=0}^{2} x_m^{(2)} - \frac{1}{3} \sum_{m=0}^{2} x_m^{(1)} \right)}_{\text{Inter-centroid displacement}}. \tag{56}$$

$d(X)$ is positive if the centroid of plane 2 lies on the side of plane 1 pointed to by its normal vector, and negative otherwise. The magnitude is the perpendicular distance between the two centroids along the normal direction of the first plane. We evaluate $d(X)$ using three pairs of bond-forming atoms (plane 1: dimethylfulvene atoms 15, 13, 24; plane 2: tropone atoms 2, 3, 7 in Fig. 6b) and apply a lower harmonic wall restraint $U_{\mathrm{plane}}(X) = \frac{1}{2}\kappa[-d(X)]_+^2$ with $\kappa = 100$ eV/Å$^2$ to enforce $d(X) > 0$.

Finally, for both WT-ASBS and WTMetaD, the no-bond restraint in Eq. (15) is applied to atom pairs C6–C24, C3–C18, and O8–C19 to suppress sampling of irrelevant reactive pathways.

**Pretraining data**   The system was propagated with a Langevin thermostat at 300 K, a 1 fs timestep, and a friction coefficient of 1 ps$^{-1}$. Initial velocities were assigned from a Maxwell–Boltzmann distribution and the total momentum was removed. After 2 ps of equilibration (2000 steps), production dynamics were carried out for 10 ps (10k steps), saving configurations every 10 fs. In total, pretraining data generation required $1.2 \times 10^4$ sequential energy evaluations.

**WTMetaD simulations**   We performed WTMetaD simulations using $(s_1, s_2)$ as the CVs. The dynamics were propagated with a Langevin thermostat at 300 K, a 0.5 fs timestep, and a friction coefficient of 10 ps$^{-1}$. Initial velocities were assigned from a Maxwell–Boltzmann distribution, and the total momentum was removed. Following a 1 ps equilibration at 300 K, Gaussian biases were

deposited every 250 steps with widths of 0.05 for both CVs. The biasing scheme used an initial Gaussian height of 0.02 eV, a bias factor of 30, and a grid spanning $[0.0, 3.0] \times [-1.0, 1.0]$. Each production run was 0.8 ns in length, and simulations were repeated four times to ensure statistical reliability, yielding $6.4 \times 10^6$ energy evaluations.

# E    ABLATION STUDIES

## E.1    WELL-TEMPERED BIAS UPDATE

Here, we examine how the well-tempered bias hyperparameters (Gaussian height $h$, width $\sigma$, and bias factor $\gamma$) influence bias deposition, the intermediate sample distribution, and the convergence of free energy differences, using alanine dipeptide as a representative example.

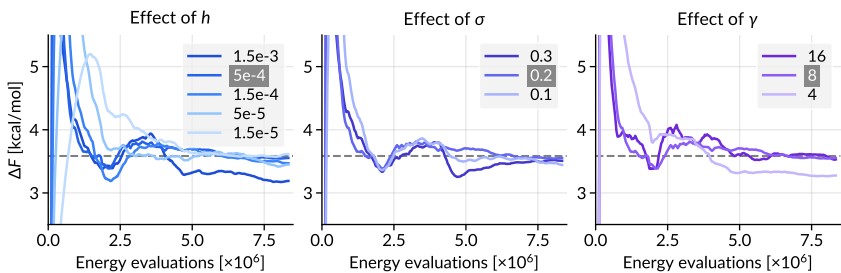

Figure 7: **Ablation of well-tempered bias updates on free energy convergence.** The free energy difference $\Delta F$ between two states of alanine dipeptide is shown as a function of the number of energy evaluations during training. Each panel varies $h$, $\sigma$, and $\gamma$, and the default values used in the main text are indicated in gray background. $h$ and $\sigma$ are in units of eV and radians, respectively.

The convergence of $\Delta F$ shown in Fig. 7 across different well-tempered bias hyperparameters indicates that the method is robust over a broad range of settings. For the Gaussian height, values from $h = 1.5 \times 10^{-5}$ to $5 \times 10^{-4}$ eV yield nearly identical converged results despite spanning almost a factor of 30, while only the largest value, $h = 1.5 \times 10^{-3}$ eV, shows a slight deviation. As seen in Fig. 8, increasing $h$ enhances early exploration of new modes and leads to a faster initial approach to the target $\Delta F$, but excessively large $h$ can impair final convergence. We note that, across different experiments, $h$ values on the order of $10^{-4}$ eV perform well. This is roughly two orders of magnitude smaller than the $h$ values commonly used in standard WTMetaD simulations, since multiple Gaussians are deposited simultaneously during WT-ASBS training.

For $\sigma$, larger values promote faster initial exploration by producing a broader bias early in training (Fig. 9), while their impact on $\Delta F$ convergence is comparatively modest (Fig. 7). For $\gamma$, we observe that a small value such as $\gamma = 4$ leads to much slower convergence (Fig. 7) and reduced exploration (Fig. 10), as the effective temperature along the CVs is lower. For both $\sigma$ and $\gamma$, we observed that values commonly used in WTMetaD simulations provide good performance.

## E.2    TRAINING MECHANISM

Now we perform analogous ablations on two training mechanisms central to the practical implementation of WT-ASBS: the replay buffer and data-based pretraining. Although several replay buffer hyperparameters, such as AM/CM buffer sizes, the number of buffer samples per epoch, and the number of gradient updates between buffer refreshes, can influence training dynamics, we use the number of AM gradient updates (Algorithm 2, $L$ in line 8) as a representative control variable to assess the effect of the replay buffer. Because the replay buffer update is coupled to the bias update, increasing $L$ makes both updates less frequent.

First, the $\Delta F$ convergence in Fig. 11 shows that the replay buffer has a substantial impact on training efficiency: using only a small number of AM steps per replay buffer update slows convergence significantly. The bias deposition patterns in Fig. 12 further indicate that overly frequent buffer and bias updates prevent the sampler parameters from keeping pace with the evolving bias, causing excessive bias to accumulate in the initial basin and leading to slow exploration of the next state. This

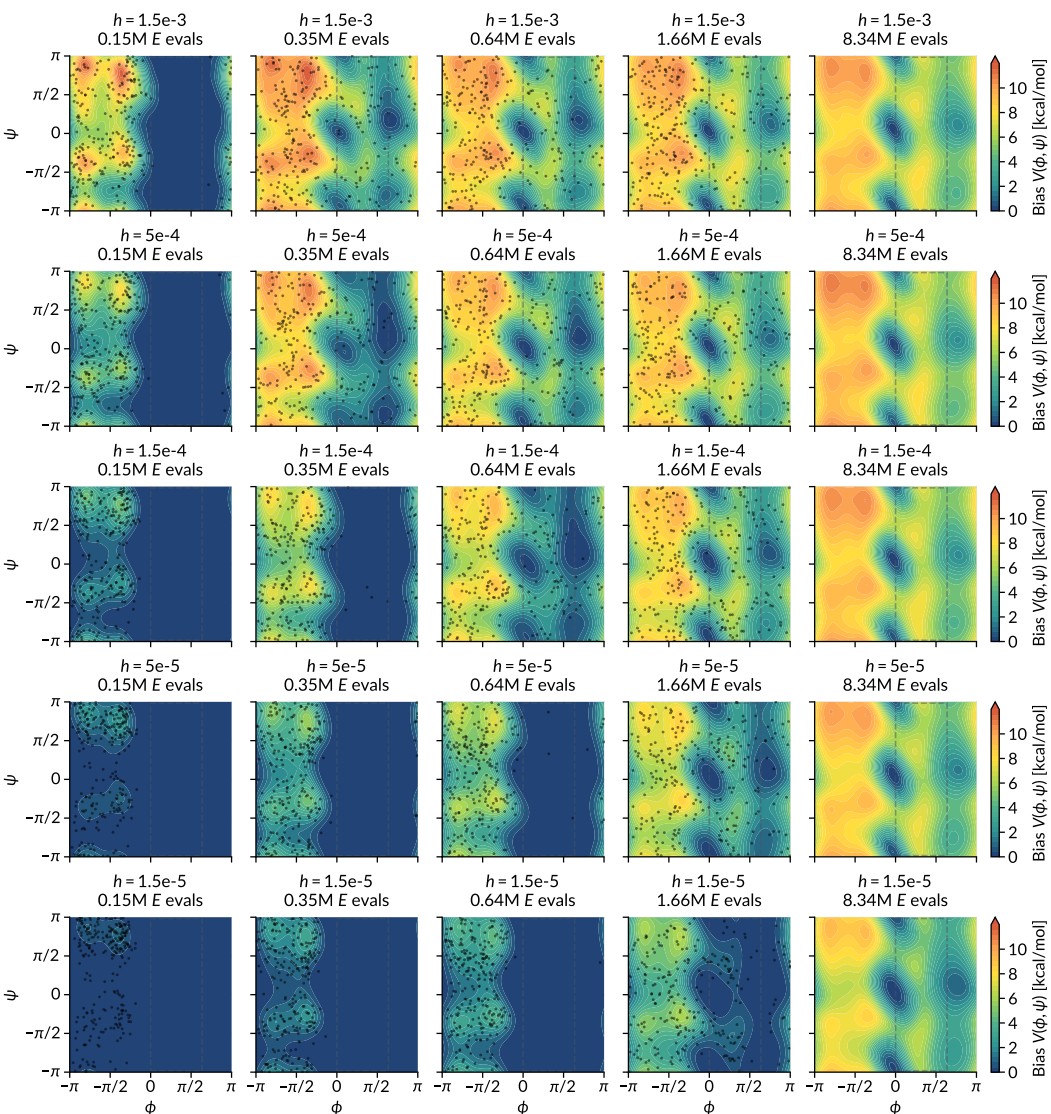

Figure 8: **Bias deposition and projected samples during training for different** $h$ **values.** Each row shows, for selected $h$, how the deposited bias $V(\phi, \psi)$ (shifted to have a minimum of zero) evolves with increasing numbers of energy evaluations, along with the samples produced by the process at the corresponding training step, projected onto the $\phi$–$\psi$ space. $h$ values are reported in units of eV.

illustrates that sufficient parameter updates between buffer updates help bridge the gap between the *ideal* two-time-scale WT-ASBS in Algorithm 1, where ASBS training converges at each fixed bias, and the *practical* WT-ASBS in Algorithm 2, where parameter and bias updates proceed concurrently. The replay mechanism must therefore provide enough time for parameter updates to keep up with bias deposition.

Next, Fig. 11 shows that $\Delta F$ convergence is substantially faster when training begins from localized pretraining. The sample distributions in Fig. 13 clarify this behavior: without pretraining, the model must identify low-energy modes from scratch, which requires a large number of energy gradient evaluations during the early stages of training. While the well-tempered bias update can converge to the correct PMF under mild conditions regardless of the initial sampling distribution, these results show that it is beneficial to exploit the initial structure of the chemical system and the efficiency of local, dynamics-based exploration.

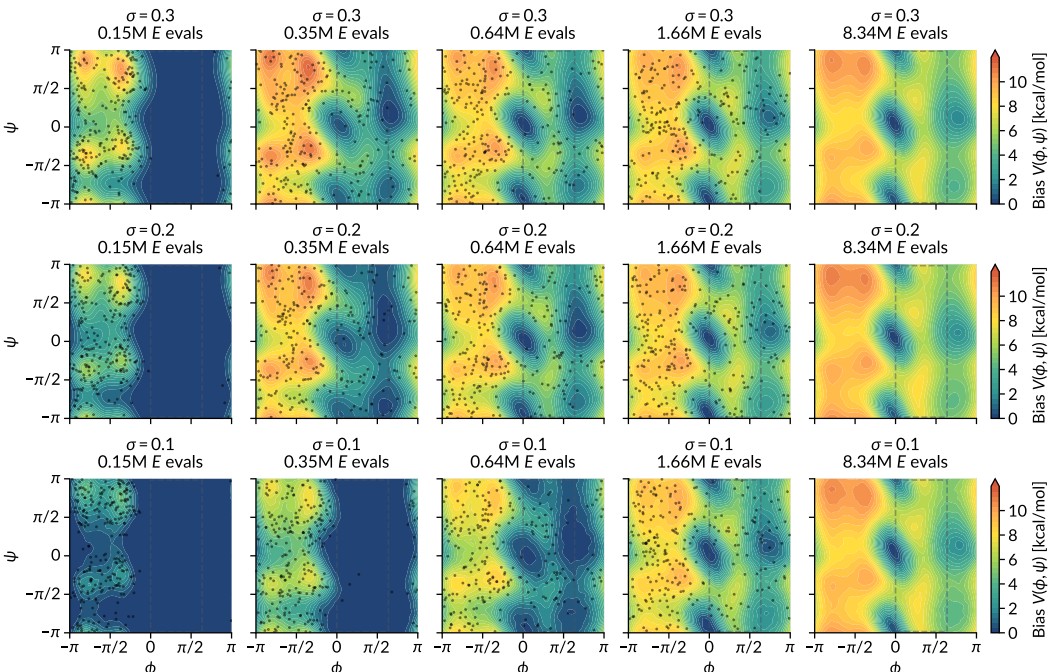

Figure 9: **Bias deposition and projected samples during training for different $\sigma$ values.** Each row shows, for selected $\sigma$, how the deposited bias $V(\phi, \psi)$ (shifted to have a minimum of zero) evolves with increasing numbers of energy evaluations, along with the samples produced by the process at the corresponding training step, projected onto the $\phi$–$\psi$ space.

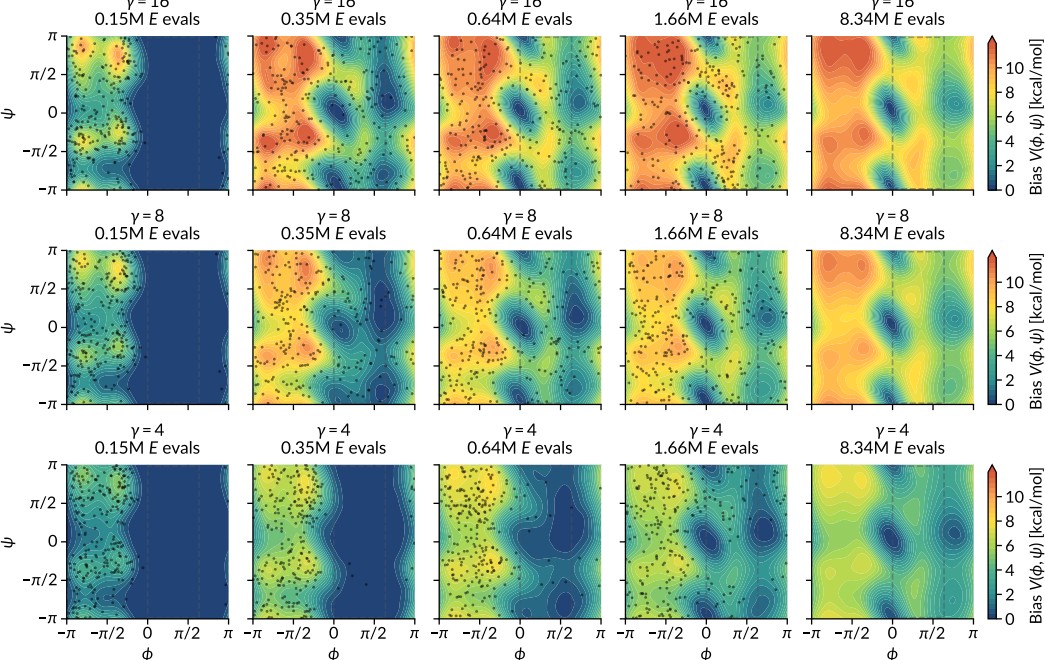

Figure 10: **Bias deposition and projected samples during training for different $\gamma$ values.** Each row shows, for selected $\gamma$, how the deposited bias $V(\phi, \psi)$ (shifted to have a minimum of zero) evolves with increasing numbers of energy evaluations, along with the samples produced by the process at the corresponding training step, projected onto the $\phi$–$\psi$ space.

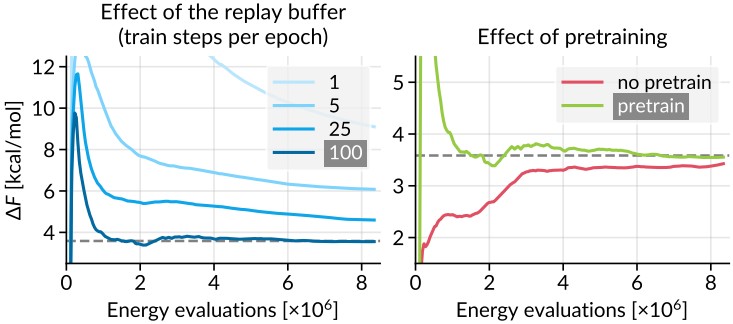

Figure 11: **Ablation of training mechanisms on free energy convergence.** The free energy difference $\Delta F$ between two states of alanine dipeptide is shown as a function of the number of energy evaluations during training. Each panel varies the number of steps per epoch (replay buffer update frequency) and whether pretraining is used.

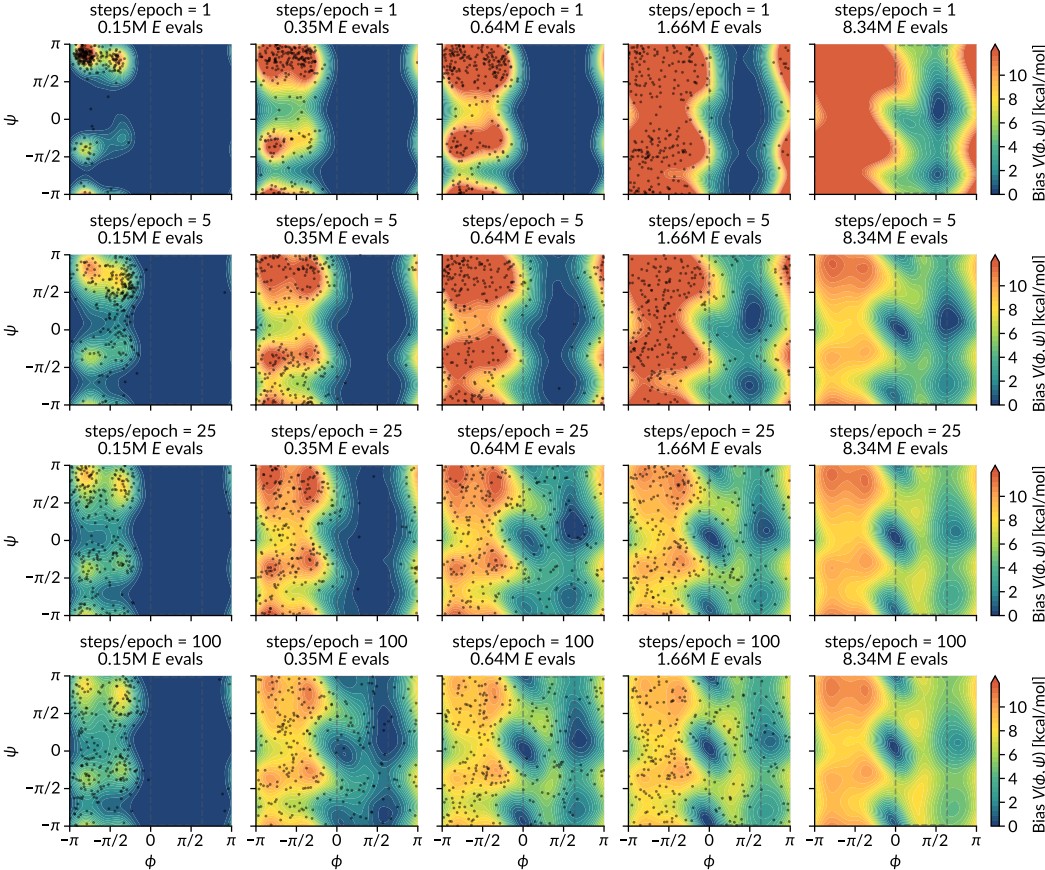

Figure 12: **Bias deposition and projected samples during training for different number of AM steps per epoch.** Each row shows, for selected number of steps per epoch, how the deposited bias $V(\phi, \psi)$ (shifted to have a minimum of zero) evolves with increasing numbers of energy evaluations, along with the samples produced by the process at the corresponding training step, projected onto the $\phi$–$\psi$ space.

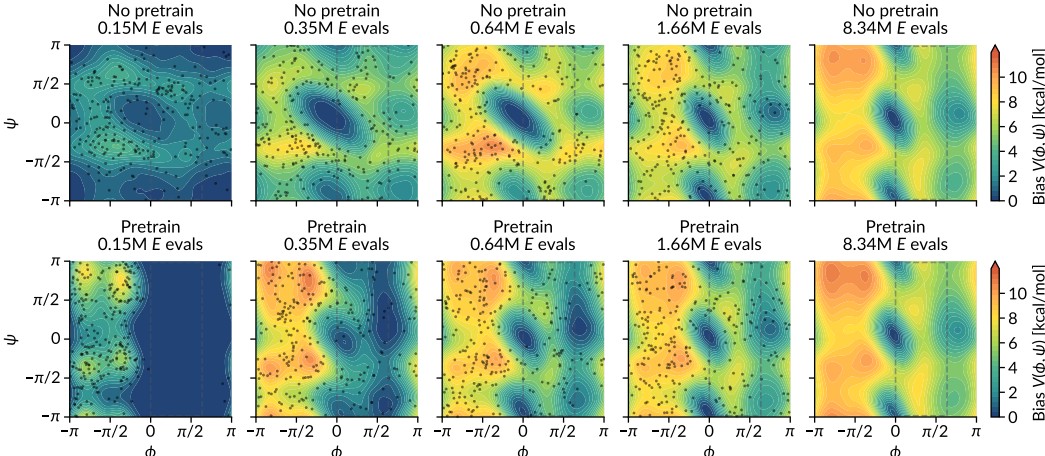

Figure 13: **Bias deposition and projected samples during training with or without pretraining.** The upper row shows results without pretraining and the lower row shows results with pretraining. Each row illustrates how the deposited bias $V(\phi, \psi)$ (shifted to have a minimum of zero) evolves with increasing numbers of energy evaluations, along with the samples produced by the process at the corresponding training step, projected onto the $\phi$–$\psi$ space.

## F  ADDITIONAL RESULTS

### F.1  ALANINE DIPEPTIDE

In this section, we extend the alanine dipeptide sampling comparison from the main text to standard distributional metrics following Liu et al. (2025). Specifically, we report KL divergence for 1-D marginals of torsion angles, as well as $\mathcal{W}_2$ and Jensen–Shannon divergence (JSD) for the joint distribution of backbone torsions $(\phi, \psi)$. Results for previous diffusion samplers using internal coordinate representations (bond distances, angles, and torsions) and for models trained in this work using Cartesian coordinates are summarized in Table 2. We used $10^6$ samples to compute $D_{\text{KL}}$ and JSD, and randomly selected $10^4$ samples for $\mathcal{W}_2$ (with weighted random sampling for WT-ASBS).

Table 2: **Distributional metrics for alanine dipeptide.** Comparison of diffusion samplers applied in internal or Cartesian coordinate representations of alanine dipeptide sampling. The KL divergence ($D_{\text{KL}}$) of five 1-D torsion angle marginals and the Wasserstein-2 distance ($\mathcal{W}_2$) and JSD on joint backbone torsion distribution are reported. $^*$Results from Liu et al. (2025). $^\dagger$Result from Choi et al. (2025). $^\ddagger$Reweighted samples are used.

| Method | $D_{\text{KL}}$ on 1-D marginal ($\downarrow$) | | | | | $(\phi, \psi)$ joint | |
| --- | --- | --- | --- | --- | --- | --- | --- |
| | $\phi$ | $\psi$ | $\gamma_1$ | $\gamma_2$ | $\gamma_3$ | $\mathcal{W}_2$ ($\downarrow$) | JSD ($\downarrow$) |
| *Internal coordinates* | | | | | | | |
| PIS (Zhang & Chen, 2021)$^*$ | $0.05_{\pm 0.03}$ | $0.38_{\pm 0.49}$ | $5.61_{\pm 1.24}$ | $4.49_{\pm 0.03}$ | $4.60_{\pm 0.03}$ | $1.27_{\pm 1.19}$ | – |
| DDS (Vargas et al., 2023a)$^*$ | $0.03_{\pm 0.01}$ | $0.16_{\pm 0.07}$ | $2.44_{\pm 0.96}$ | $0.03_{\pm 0.00}$ | $0.03_{\pm 0.00}$ | $0.68_{\pm 0.09}$ | – |
| AS (Havens et al., 2025)$^*$ | $0.09_{\pm 0.09}$ | $0.04_{\pm 0.04}$ | $0.17_{\pm 0.17}$ | $0.56_{\pm 0.09}$ | $0.51_{\pm 0.06}$ | $0.65_{\pm 0.52}$ | – |
| NAAS (Choi et al., 2025)$^\dagger$ | $0.26$ | $0.24$ | $0.27$ | $0.13$ | $0.16$ | – | – |
| ASBS (Liu et al., 2025)$^*$ | $0.02_{\pm 0.00}$ | $0.01_{\pm 0.00}$ | $0.03_{\pm 0.01}$ | $0.02_{\pm 0.00}$ | $0.02_{\pm 0.00}$ | $0.25_{\pm 0.01}$ | – |
| *Cartesian coordinates* | | | | | | | |
| ASBS (Liu et al., 2025) | $0.14_{\pm 0.03}$ | $0.09_{\pm 0.03}$ | $0.02_{\pm 0.00}$ | $0.00_{\pm 0.00}$ | $0.00_{\pm 0.00}$ | $0.45_{\pm 0.01}$ | $0.014_{\pm 0.003}$ |
| WT-ASBS (this work)$^\ddagger$ | $0.02_{\pm 0.00}$ | $0.04_{\pm 0.01}$ | $0.02_{\pm 0.00}$ | $0.00_{\pm 0.00}$ | $0.00_{\pm 0.00}$ | $0.43_{\pm 0.02}$ | $0.005_{\pm 0.001}$ |

Table 2 shows that WT-ASBS performs well on all metrics, with similar $\mathcal{W}_2$ performance to baseline ASBS (Cartesian) and slightly worse than reference ASBS (internal coordinates). However, the Ramachandran plots in Fig. 14 reveal that WT-ASBS better reproduces the overall density, while the other two methods either collapse into highly populated regions or leak into sparsely populated ones. This discrepancy arises because the metrics in Table 2 assume equal weight for all samples.

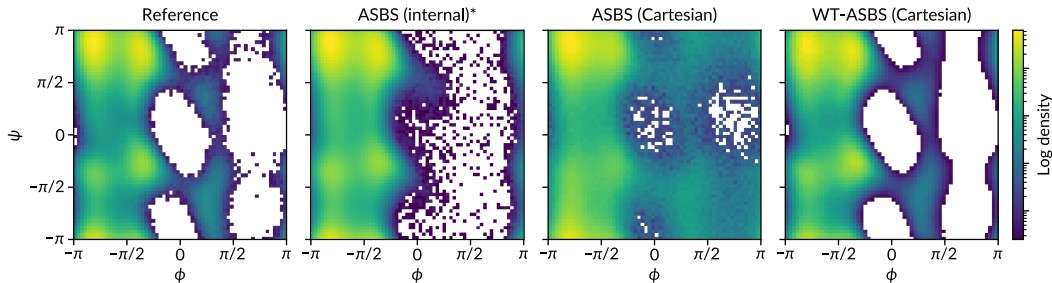

Figure 14: **Ramachandran plots for alanine dipeptide.** Comparison of backbone torsion distributions $(\phi, \psi)$ in log densities, for reference, ASBS (internal coordinates), ASBS and WT-ASBS (Cartesian coordinates), as in Table 2. All three methods reproduce the more populated regions reasonably well, but their behavior in less populated regions differs substantially. Negligibly low-density regions are omitted for visual clarity. *Taken from Liu et al. (2025).

WT-ASBS, designed to satisfy Requirement 3, incorporates weighted sampling to achieve balanced efficiency across all relevant modes. Since Boltzmann populations depend *exponentially* on free energy, focusing on more populated regions can yield good metrics without capturing the full distribution. Therefore, we emphasize evaluations more closely aligned with the design choices discussed in the main text.

## F.2 ALANINE TETRAPEPTIDE

In Fig. 15, we show the distribution of pretraining data and compare the PMF obtained by reweighting (Eq. (1)) for WT-ASBS, the bias (Remark 3.1) for WTMetaD and WT-ASBS, and the reference PMF. The populations within each mode are well reproduced, while biased sampling methods (WT-ASBS and WTMetaD) capture the energetic features of the transition region between modes. This region is inaccessible even with a huge number of unbiased samples ($5 \times 10^7$ for the reference), underscoring the necessity of biased sampling.

## F.3 SAMPLING REACTIVE ENERGY LANDSCAPES

To quantify the convergence of PMFs in Fig. 5, we define the PMF MAE between a PMF $F(s)$ and its reference $F_{\text{ref}}(s)$, both shifted to have minimum zero, as

$$\text{MAE}[F, F_{\text{ref}}] = \frac{\min_c \int_{\mathcal{S}} |F(s) - F_{\text{ref}}(s) + c| \, \mathbf{1}[F_{\text{ref}}(s) < F_{\text{thres}}] \, \mathrm{d}s}{\int_{\mathcal{S}} \mathbf{1}[F_{\text{ref}}(s) < F_{\text{thres}}] \, \mathrm{d}s}, \tag{57}$$

i.e., the average absolute deviation (up to an optimal constant shift $c$) over regions of CV space where $F_{\text{ref}}(s)$ is below the threshold $F_{\text{thres}}$. We used $F_{\text{thres}}$ of 30 kcal/mol for the $S_N 2$ reaction, and 45 kcal/mol for the post-TS bifurcation reaction.

We compare PMFs for reactions obtained with bias from WT-ASBS and WTMetaD in Figs. 16 and 17, and confirm that the final PMFs from both methods agree well. For the post-TS bifurcation, valuable chemical insights into the mechanistic understanding of reactions can be obtained from the free energy profiles. For example, while **5** is the experimentally observed product, two competing mechanisms of **3** → **4** → **5** (Paddon-Row & Warrener, 1974) vs. **3** → **5** (Houk et al., 1970) have been proposed. The 2-D PMF reveals that the initial reactive flux is biased toward product **4**, but the deeper free energy basin of **5** makes it the thermodynamically favored product, consistent with previous findings that **4** forms transiently and subsequently converts to **5** (Yu et al., 2017).

## F.4 ML COLLECTIVE VARIABLES

To illustrate the generalizability of the method to non–hand-crafted CVs, we present initial experiments coupling WT-ASBS with an ML CV learned automatically from unbiased samples of the metastable states of alanine dipeptide.

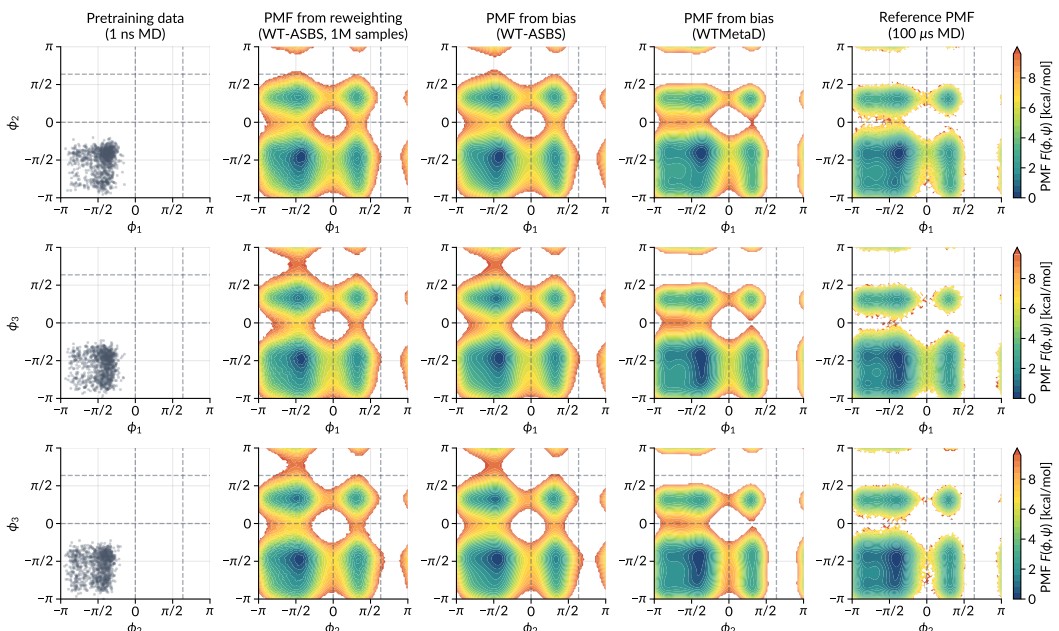

Figure 15: **Alanine tetrapeptide PMFs.** Columns show pretraining data, PMFs from reweighting (WT-ASBS), PMFs from WT-ASBS and WTMetaD biases, and the reference PMF. Rows correspond to different torsion angle pairs $(\phi_1, \phi_2)$, $(\phi_1, \phi_3)$, and $(\phi_2, \phi_3)$. Dotted lines at $\phi_i = 0$ and 2 mark the mode boundaries. All method reproduces populations within metastable modes, while biased sampling (WT-ASBS and WTMetaD) captures transition regions inaccessible to unbiased sampling.

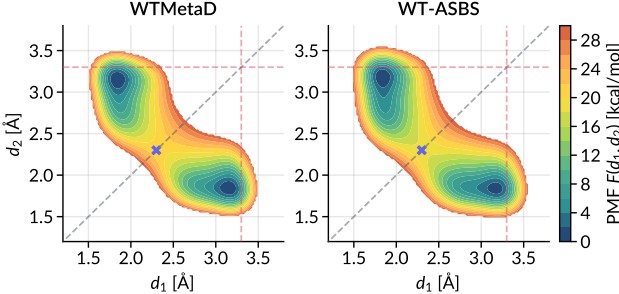

Figure 16: **S$_N$2 reaction PMFs.** Comparison of PMFs obtained from WT-ASBS and WTMetaD biases. Red dotted lines mark regions influenced by restraints, gray dotted lines denote the $d_1 = d_2$ symmetry lines, and the purple cross indicates the transition state location.

Starting from 5,000 unbiased configurations collected from each state (Fig. 18a), we trained a Deep-TDA (Trizio & Parrinello, 2021) CV that maps the samples to a one-dimensional space in which each state corresponds to a single Gaussian distribution. We used the default settings of the mlcolvar package (Bonati et al., 2023), employing a multilayer perceptron CV model that takes pairwise heavy-atom distances as inputs and includes hidden layers of sizes 24 and 12. We assign states 1 and 2 to Gaussians $\mathcal{N}(-7.0, 1.0^2)$ and $\mathcal{N}(7.0, 1.0^2)$, respectively. The resulting ML CV values for long unbiased reference configurations (including transitions) are shown in Fig. 18b: configurations from the two metastable states cluster around $-7$ and 7, while transition configurations lie smoothly between them. We then perform WT-ASBS training using the same settings as in the main-text alanine dipeptide experiment, except for a smaller Gaussian height $h = 1 \times 10^{-5}$ eV to account for the reduced CV dimensionality.

The final 1-D PMF along the CV in Fig. 18c shows good agreement with the reference PMF, and $\Delta F$ approaches the reference value smoothly (Fig. 18f). The 2-D PMF on $(\phi, \psi)$ obtained by reweighting 1M samples with their bias values (Fig. 18d) also agrees reasonably well with the reference 2-D PMF

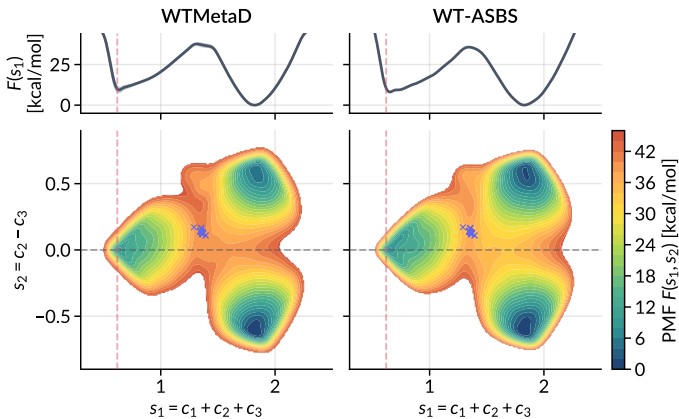

Figure 17: **Post-TS bifurcation reaction PMFs.** Comparison of 1-D PMF $F(s_1)$ and 2-D PMF $F(s_1, s_2)$ obtained from WT-ASBS and WTMetaD biases. Red dotted lines mark regions influenced by restraints, gray dotted lines denote the $c_2 = c_3$ symmetry lines, and the purple cross indicates the transition state locations.

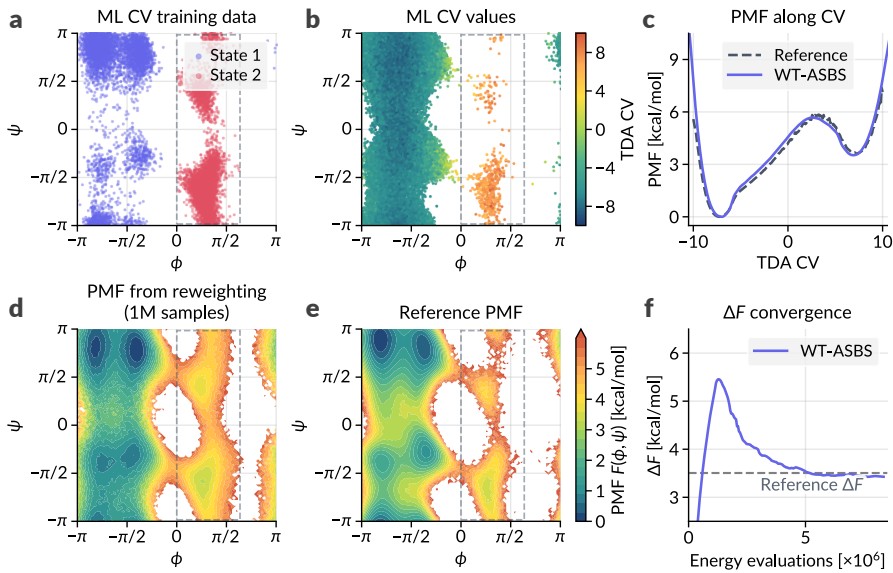

Figure 18: **WT-ASBS for alanine dipeptide with ML CV.** (a) CV training data from each state in the $(\phi, \psi)$ space. (b) Trained ML CV values for reference equilibrium configurations. (c) One-dimensional PMF along the Deep-TDA CV obtained with WT-ASBS and from reference unbiased configurations. (d, e) Two-dimensional PMF on $(\phi, \psi)$ from (d) WT-ASBS with reweighting and (e) reference unbiased configurations. (f) Convergence of $\Delta F$ between the two states for WT-ASBS as a function of the number of energy evaluations.

(Fig. 18e), though the modes appear broadened along the $\psi$ direction, which is less well resolved by the ML CV. This behavior is consistent with the difference between baseline ASBS and WT-ASBS (Fig. 14), where WT-based weights help recover structure in sparsely sampled regions by downweighting configurations that are distinguishable in the CV space. These preliminary results suggest that WT-ASBS can be combined with ML CV pipelines to enable enhanced exploration when a good CV is not known a priori.

## G    LARGE LANGUAGE MODEL STATEMENT

We made limited use of large language models (LLMs) to assist with language refinement and proofreading. No content, ideas, or analyses were produced by these tools.

