# OpenReview forum: "Enhancing Diffusion-Based Sampling with Molecular Collective Variables"
_ICLR.cc/2026/Conference — ICLR 2026 Poster_

### Official Review · Reviewer_mrdW · 2025-10-29

**Soundness:** 4
**Presentation:** 4
**Contribution:** 3
**Rating:** 6
**Confidence:** 4

**Summary:**

This paper introduces WT-ASBS, a diffusion-based generative sampler enhanced by biasing along  CVs, inspired by WTMetaD.

The method replaces the local MD step in WTMetaD with a diffusion-based neural sampler (ASBS) that learns to generate molecular configurations directly from the potential energy function. A repulsive potential is iteratively constructed in CV space to encourage exploration of new regions and flatten free-energy barriers, with subsequent reweighting to recover the unbiased Boltzmann ensemble. Experiments demonstrate that WT-ASBS improves sampling efficiency and mode discovery for molecular systems.

**Strengths:**

1. The idea of integrating diffusion-based neural samplers with enhanced-sampling techniques from molecular simulation is elegant and promising. While both ASBS and WTMetaD exist independently, their combination is novel and creates a bridge between machine-learning-based and physics-based sampling paradigms.

2. The methodology is technically well-founded, with a clear derivation and an explicit algorithmic description (Algorithm 1). The paper includes a convergence proposition showing that the bias potential approaches the well-tempered limit, ensuring theoretical consistency.

3. The paper is clearly written, logically organized, and visually well-supported by figures. The workflow in Fig. 1 and the schematic diagrams for peptide systems help the reader follow the training and bias-updating procedure.

4. From an application perspective, the work demonstrates that diffusion-based samplers can efficiently handle physically meaningful molecular systems. The results on peptides and reactive landscapes show large efficiency gains over WTMetaD. This points toward real practical potential in molecular modeling.

**Weaknesses:**

1. The method mainly replaces the MD propagation in WTMetaD with ASBS without introducing new learning objectives or architectures. The innovation lies more in application and system integration than in core ML algorithmic development. For ICLR, where the focus is typically on advancing machine-learning methodology, the contribution might appear not so significant. The work may find a more natural home in computational chemistry or physics-oriented venues.

2. Algorithm 1 involves two nested loops (outer bias update over k, inner ASBS optimization over l). In addition, an iid sampling procedure is involved. However, the paper does not specify when or how to stop the iterations in practice. ASBS itself is self-consistent and lacks a single scalar loss whose decrease guarantees convergence. The absence of heuristic or empirical convergence criteria (e.g., stabilization of bias potential or free-energy estimates) makes reproducibility difficult.

3. All experiments use well-known CVs such as torsional angles or bond distances. For realistic, high-dimensional systems, CV discovery is non-trivial. It would strengthen the paper to evaluate the method with ML-discovered CVs (e.g., TICA, SPIB, or FMRC). Without this, the method’s applicability to problems where good CVs are not known remains uncertain.

4. Boltzmann Generators and related approaches can guarantee asymptotically correct estimates via MCMC or importance-sampling refinement. WT-ASBS claims reweighting via the bias but does not discuss whether such reweighting ensures unbiased expectations in the presence of neural-sampler approximation error.

5. In Fig. 3d, the free-energy mean-absolute-error of WT-ASBS is slightly larger than that of WTMetaD. It would be valuable to analyze the reason to understand when diffusion-based sampling may underperform.

6. The method’s feasibility for explicit-solvent systems remains unclear.

**Questions:**

1. How should practitioners determine that the two-loop WT-ASBS training has converged?

2. Can the authors clarify whether the proposed reweighting scheme guarantees unbiased Boltzmann expectations, or whether the residual approximation in the learned diffusion process introduces systematic bias? Could additional refinement (e.g., short MCMC runs or importance sampling) ensure asymptotic correctness similar to Boltzmann Generators?

3. Have the authors considered coupling WT-ASBS with ML-based CV discovery?

4. WT-ASBS achieves broader exploration but sometimes slightly worse free-energy accuracy (Fig. 3d). Can the authors comment on this?

5. Can the authors explore the computational feasibility for systems with explicit solvent?

---

> ### Author Response · Authors · 2025-11-25
> **Replies to Reviewer mrdW (Part 1)**
>
> We thank the reviewer for the encouraging evaluation and for dedicating time to assess our submission. Please find our point-to-point replies below.
>
> > **W1.** The method mainly replaces the MD propagation in WTMetaD with ASBS without introducing new learning objectives or architectures. The innovation lies more in application and system integration than in core ML algorithmic development. For ICLR, where the focus is typically on advancing machine-learning methodology, the contribution might appear not so significant. The work may find a more natural home in computational chemistry or physics-oriented venues.
>
> While one way to view the method is as a replacement of MD propagation in WTMetaD with ASBS, an equally important view is that WTMetaD provides the missing mechanism required to overcome a core ML challenge of ASBS and diffusion samplers: the strong mode-seeking behavior of energy-based training without access to samples from different modes. ASBS alone tends to concentrate sampling mass near already discovered low-energy regions, which can prevent systematic exploration of new modes. The well-tempered bias update introduces a principled, physics-motivated mechanism to correct this behavior by promoting controlled exploration in a low-dimensional projection of configuration space.
>
> From the ML perspective, this low-dimensional projection is not only a practical engineering choice but also a useful abstraction. By decoupling exploration and high-dimensional sampling, it creates a structured interface that would enable new algorithmic developments on different types of "enhanced" neural samplers designed to operate on the evolving biased distribution defined through the projection. We therefore see the contribution not simply as an application of an existing MD technique but as a step toward integrating principled enhanced sampling ideas into the training dynamics of high-dimensional diffusion samplers.
>
> > **W2.** Algorithm 1 involves two nested loops (outer bias update over k, inner ASBS optimization over l). In addition, an iid sampling procedure is involved. However, the paper does not specify when or how to stop the iterations in practice. ASBS itself is self-consistent and lacks a single scalar loss whose decrease guarantees convergence. The absence of heuristic or empirical convergence criteria (e.g., stabilization of bias potential or free-energy estimates) makes reproducibility difficult.
> >
> > **Q1.** How should practitioners determine that the two-loop WT-ASBS training has converged?
>
> In practice, convergence of WT-ASBS is determined using the same empirical stabilization criteria that are standard in the enhanced sampling literature, including WTMetaD. Specifically, we monitor whether (1) the newly deposited bias is small enough and (2) the estimated free energy (difference) over the chosen CVs has stabilized. These criteria are exactly the signals the reviewer mentioned and are the same diagnostics typically used to assess convergence of a well-tempered simulation. Although ASBS does not optimize a single scalar objective, the well-tempered update produces a clear notion of progress. As the bias growth slows exponentially (yhe exponential prefactor in eq. 12), the system transitions from exploration toward refinement, and both the bias landscape and $\Delta F$ estimates stabilize.
>
> > **W3.** All experiments use well-known CVs such as torsional angles or bond distances. For realistic, high-dimensional systems, CV discovery is non-trivial. It would strengthen the paper to evaluate the method with ML-discovered CVs (e.g., TICA, SPIB, or FMRC). Without this, the method’s applicability to problems where good CVs are not known remains uncertain.
> >
> > **Q3.** Have the authors considered coupling WT-ASBS with ML-based CV discovery?
>
> We agree that CV discovery is challenging for realistic systems and that it is important to assess whether WT-ASBS can be combined with ML-based CVs. To address this, we added an experiment in which the CV is not hand-designed but learned automatically (Appendix F.4). Using unbiased samples from two metastable states of alanine dipeptide, we train a one-dimensional Deep-TDA CV that separates the states. WT ASBS applied to this learned CV successfully reconstructs the 1-D PMF, yields smooth convergence of $\Delta F$, and produces a reweighted 2-D PMF in good agreement with reference data.
>
> While alanine dipeptide remains a controlled test system, this result demonstrates that WT-ASBS can operate on a learned representation and suggests a practical pipeline for more complex problems: first obtain a coarse set of metastable configurations, then learn a CV that distinguishes these modes, and finally run WT-ASBS on the learned CV. This removes the requirement that CVs be known a priori and clarifies how the method can be extended to high-dimensional systems where CV discovery is non-trivial.

---

> > ### Author Response · Authors · 2025-11-25
> > **Replies to Reviewer mrdW (Part 2)**
> >
> > > **W4.** Boltzmann Generators and related approaches can guarantee asymptotically correct estimates via MCMC or importance-sampling refinement. WT-ASBS claims reweighting via the bias but does not discuss whether such reweighting ensures unbiased expectations in the presence of neural-sampler approximation error.
> > >
> > > **Q2.** Can the authors clarify whether the proposed reweighting scheme guarantees unbiased Boltzmann expectations, or whether the residual approximation in the learned diffusion process introduces systematic bias? Could additional refinement (e.g., short MCMC runs or importance sampling) ensure asymptotic correctness similar to Boltzmann Generators?
> >
> > The reviewer is correct to point out that the reweighting via the bias is exact only when we assume the sampler is trained perfectly, and as the reviewer suggested, there are two refinement methods. The first is to use importance sampling to debias the weights induced by an imperfect sampler by using the path importance weights in ASBS (eq. 83, arXiv:2506.22565v1), where the endpoint log potential terms become tractable when we parametrize the control and corrector networks as a gradient of the corresponding SB potentials. We observed that the path weights showed large variance due to integration over stochastic terms. A more reliable option is to run short MCMC or MD starting from the sampled distribution on the final biased potential $E + V \circ \xi$ and then reweight using $V$. We have discussed this in Section 3.2 and added comments on the limitations of an imperfectly trained sampler and the high variance of the path weights.
> >
> > > **W5.** In Fig. 3d, the free-energy mean-absolute-error of WT-ASBS is slightly larger than that of WTMetaD. It would be valuable to analyze the reason to understand when diffusion-based sampling may underperform.
> > >
> > > **Q4.** WT-ASBS achieves broader exploration but sometimes slightly worse free-energy accuracy (Fig. 3d). Can the authors comment on this?
> >
> > To reproduce free energy differences accurately, both broad exploration across modes and thorough local sampling within each mode matter. Diffusion samplers are effective at generating diverse configurations globally, but MD-based local exploration can be advantageous when the system has many degrees of freedom and intra-mode motion involves barriers that are not very large. In such cases, the sampler must learn all degrees of freedom, which can reduce efficiency and accuracy. This connects to the reviewer’s point about explicit solvent systems: for scalable sampling in chemical settings, it is important to determine where global sampling (via measure transport) is needed to overcome significant energetic barriers, and where local sampling is more suitable because movement on the energy landscape is already reasonably efficient. We have incorporated this discussion in the revised manuscript (Section 4.1, p. 8--9; Section 5, p. 10).
> >
> > > **W6.** The method’s feasibility for explicit-solvent systems remains unclear.
> > >
> > > **Q5.** Can the authors explore the computational feasibility for systems with explicit solvent?
> >
> > While we have focused on implicit solvent systems (small peptides) and reactive systems in vacuum for computational efficiency, we acknowledge that conformational distributions differ significantly between explicit and implicit solvent models, and that explicit solvent modeling will be essential for practical applications.
> >
> > In principle, the method is applicable to explicit solvent systems. The main modification needed is to incorporate periodic boundary conditions in the solvated simulation box, which can be handled by modeling the base diffusion process on a torus (e.g., summing the regression target scores over periodic images). Although adding a sufficient solvation shell increases the system size by at least a few hundred atoms, the trainable components and targets scale linearly (with graph neural network and neighbor cutoffs), so training would remain feasible with a larger compute budget.
> >
> > However, we note that an important feature of explicit solvent modeling is that solvent reorganization generally has low barriers, aside from a few localized solvent-mediated interactions. For this reason, using a diffusion sampler to generate solvent and solute jointly would be less practical, because solvent degrees of freedom could be sampled very efficiently by local samplers (running MD). To support energy-based sampling that accounts for solvent effects, an important direction for future work would be to develop a strategy that separates global sampling of the solute from local sampling of the solvent.
> >
> > ---
> >
> > Thank you,
> >
> > Submission13338 authors

---

> > > ### Comment · Reviewer_mrdW · 2025-11-27
> > >
> > > Thank you for the detailed rebuttal and clarifications. After going through the authors’ response, I will maintain my current score (still relatively positive). The rebuttal helps address several earlier concerns, and overall I believe the paper has potential impact.
> > >
> > > I would appreciate clarification on two remaining points, which I think would substantially improve the camera-ready version and help readers better understand the practical value of the proposed framework:
> > >
> > > 1. Based on my understanding, one major motivation of the paper is to accelerate simulations by identifying low-dimensional reaction coordinates, which is indeed a key topic in molecular dynamics. However, are there other fields where similar needs exist—i.e., where identifying low-dimensional latent variables can significantly accelerate sampling or long-time simulation? Listing concrete examples would better demonstrate the paper’s broader relevance and potential applicability.
> > >
> > > 2. The method involves multiple nested loops and iterative updates. In practice, how should users determine when the algorithm has converged? And how should the number of iterations at each level be chosen? Providing more actionable rules of thumb, or stopping criteria would be extremely helpful for practitioners who want to apply the proposed method.

---

> > > > ### Author Response · Authors · 2025-12-01
> > > > **Additional replies to Reviewer mrdW**
> > > >
> > > > Thank you again for your careful reading of our work, your positive evaluation, and for highlighting two points for improving the manuscript. We have updated the current version accordingly, and we summarize the changes below:
> > > >
> > > > > 1. Based on my understanding, one major motivation of the paper is to accelerate simulations by identifying low-dimensional reaction coordinates, which is indeed a key topic in molecular dynamics. However, are there other fields where similar needs exist—i.e., where identifying low-dimensional latent variables can significantly accelerate sampling or long-time simulation? Listing concrete examples would better demonstrate the paper’s broader relevance and potential applicability.
> > > >
> > > > We agree that making the connection to other application domains more explicit will clarify the framework's broader impact. There are a few cases where leveraging a low-dimensional latent descriptor accelerates the sampling outside molecular configurations:
> > > >
> > > > - Swendsen-Wang sampling for Ising/Potts models: Cluster labels are introduced as auxiliary latent variables. These latent cluster variables define nonlocal moves that jointly flip large, correlated regions, dramatically accelerating mixing compared to single-spin updates in discrete lattice systems.
> > > > - Wang-Landau sampling: One effectively learns a low-dimensional bias as a function of the energy (an approximate density of states). This one-dimensional latent "energy coordinate" is iteratively adjusted to flatten the energy histogram, accelerating exploration of configuration space and convergence of thermodynamic quantities.
> > > > - Rao-Blackwellized particle filters (RBPFs) for dynamic Bayesian networks: Part of the state is represented by a low-dimensional latent process that can be marginalized analytically, while only a reduced set of variables is sampled. This structured latent representation improves the efficiency and stability of long-time sequential inference.
> > > >
> > > > We added a final remark to the Discussion regarding this: "Finally, although this work focuses on sampling molecular configurations, low-dimensional latent variables have proven useful in discrete lattice sampling (Swendsen & Wang, 1987; Wang & Landau, 2001) and dynamic Bayesian networks (Doucet et al., 2000), suggesting that extending the WT-ASBS-like approach beyond molecular systems would be a fruitful direction."
> > > >
> > > > We would also like to note that, while this connects our work within the broader area of ML-based sampling, its primary focus is molecular sampling, which is already a major topic at ICLR. In particular, this is reflected in our chosen Primary Area, “applications to physical sciences (physics, chemistry, biology, etc.),” and we hope that the contribution can be evaluated in light of this field designation.
> > > >
> > > > > 2. The method involves multiple nested loops and iterative updates. In practice, how should users determine when the algorithm has converged? And how should the number of iterations at each level be chosen? Providing more actionable rules of thumb, or stopping criteria would be extremely helpful for practitioners who want to apply the proposed method.
> > > >
> > > > We appreciate your request for more actionable guidance on convergence and iteration choices. While we have explained the indicators for convergence of outer-loop bias updates in the reply to **W2**/**Q1**, we have added a more detailed and practitioner-oriented discussion in Appendix C.4, including inner loop AM/BM training:
> > > >
> > > > - The number of gradient updates $L$ is chosen so that the controller parameters can track the outer-loop bias updates, as validated in the ablation studies in Appendix E.2.
> > > > - The AM and CM epochs per stage ($M_\text{adj}, M_\text{crt}$) follow prior work (Liu et al., 2025) and are selected so that the corresponding losses stabilize within each stage.
> > > > - The number of alternating stages $K$ is determined by monitoring convergence of the bias or PMF: in practice, we stop when (i) newly deposited bias along the selected CVs becomes negligible over several stages, or (ii) estimated free energy differences along these CVs stabilize within a chosen tolerance.
> > > >
> > > > ---
> > > >
> > > > Thank you,
> > > >
> > > > Submission13338 authors

---

### Official Review · Reviewer_2tNT · 2025-10-31

**Soundness:** 4
**Presentation:** 3
**Contribution:** 3
**Rating:** 6
**Confidence:** 3

**Summary:**

This paper proposes WT-ASBS (Well-Tempered Adjoint Schrödinger Bridge Sampler), a method that combines a neural diffusion-based sampler (ASBS) with a well-tempered bias potential defined in collective-variable (CV) space. The goal is to accelerate exploration in molecular systems by gradually flattening the free-energy landscape while retaining thermodynamic consistency through reweighting.

The authors claim that this hybrid design achieves faster and broader sampling compared to standard ASBS and conventional enhanced sampling methods such as WTMetaD. Empirical results on alanine dipeptide, alanine tetrapeptide, and chemical reaction benchmarks suggest improved exploration and free-energy reconstruction.

**Strengths:**

**Conceptually innovative hybridization.**
The integration of Schrödinger bridge–based diffusion sampling with a well-tempered metadynamics bias is novel. It bridges probabilistic transport and enhanced sampling, offering a unified view of energy-driven exploration in molecular systems.

**Clear motivation from sampling efficiency.**
The work addresses a concrete challenge in molecular generative modeling — slow mode discovery and poor coverage of rare events — and proposes a pragmatic biasing mechanism that is both simple and theoretically interpretable.

**Transparent discussion of limitations.**
The paper openly discusses the non-amortized bias, potential generalization issues, and differences from traditional WTMetaD, which reflects scientific maturity and honesty.

**Weaknesses:**

**1. Conceptual tension between learned sampler and non-learnable bias.**
Although the method is presented as a neural diffusion sampler, the core bias $V_{WT}$ is manually accumulated (non-neural). This design choice blurs whether the sampler truly learns the target distribution or merely follows a hand-crafted MetaD bias.

**2. Lack of fair and direct comparison with ASBS.**
Since WT-ASBS is an ASBS variant augmented with a well-tempered bias, the most meaningful benchmark is ASBS itself under identical conditions. However, ASBS only appears in Fig. 3c (explored states) and is not evaluated for PMF accuracy, making the incremental benefit of WT unclear.

**3. # of force-calls inconsistency between figures.**
Figure 3.c shows up to $\approx$2 M evaluations, while Figure 3.d extends to $\approx$40 M. Because WT-ASBS reaches chemical accuracy after $\approx$4–5 M evaluations, truncating Fig. 3c prevents fair comparison at later stages.

**4. Limited diffusion-based baselines.**
Comparisons are restricted to WTMetaD, which mainly highlights conceptual differences rather than validating the WT mechanism across diffusion methods. Adding at least one diffusion-based baseline (annealed, energy-guided, or score-guided) would strengthen the evaluation.

**5. Scalability and consistency.**
WT-ASBS excels on alanine dipeptide (Figure 2.f) but underperforms WTMetaD on tetrapeptide (Figure 3.d), suggesting potential sensitivity to CV dimensionality or generalizability to complex system.

**6. Lack of statistical robustness.**
The reported results appear to be based on a single simulation run for WT-ASBS. Given the stochastic nature of both diffusion and metadynamics sampling, multiple independent runs (with mean and variance) are crucial to assess convergence stability and reproducibility. Without them, it is difficult to judge whether the reported trajectories and PMF curves reflect consistent behavior or a favorable random seed.

**Questions:**

(Q1) If $V_{WT}(\xi)$ is not parameterized by a neural network, how does the model ensure that the learned sampler contributes beyond a standard MetaD bias? Could the authors clarify whether WT-ASBS should be interpreted as (a) a neural sampler guided by an analytic bias, or (b) a learnable bias parameterized through the neural transport model?

(Q2) In Figure 8, the reweighted and bias-derived PMFs are nearly identical, suggesting that the sampling in CV space is almost uniform. If so, what ensures that the sampled structures remain physically valid? Could the authors provide representative molecular structures sampled from different regions of the CV space (e.g., high- vs low-free-energy regions) to illustrate what kind of configurations the model actually produces? This would clarify whether the method truly explores meaningful conformations or simply performs uniform random exploration over CVs.

(Q3) Could you extend Figure 3.c to the same energy-evaluation range ($\approx$10–40 M) as Figure 3.d to show late-stage exploration and better quantify the WT bias’s long-term effects? Please include an ASBS PMF accuracy curve for Ala2 system.

(Q4) Could you summarize cost–benefit metrics (energy evaluations, wall-clock time, ESS, PMF error) across ASBS, WT-ASBS, and WTMetaD for standardized comparison?

(Q5) ASBS’s slow exploration (Figure 3.c) might result from strong pre-training confinement near the initial mode (marked * in Figure 3.b).
Have you explored reducing pretraining strength or fine-tuning its duration to encourage broader exploration?

(Q6) When computing the free-energy difference, which states or basins were used? Were they predefined minima or clusters in CV space?
Clarifying this would help interpret the reported PMF results.

(Q7) Are all reported curves from single runs, or averaged over multiple simulations? If single, could you comment on the run-to-run variability? Showing variance bands or repeated trials would help establish statistical reliability.

---

> ### Author Response · Authors · 2025-11-25
> **Replies to Reviewer 2tNT (Part 1)**
>
> We thank the reviewer for the positive remarks and for their time and effort in evaluating our manuscript. Please find our point-to-point replies below.
>
> > **W1. Conceptual tension between learned sampler and non-learnable bias.** Although the method is presented as a neural diffusion sampler, the core bias $V_{WT}$ is manually accumulated (non-neural). This design choice blurs whether the sampler truly learns the target distribution or merely follows a hand-crafted MetaD bias.
> >
> > **Q1.** If $V_{WT}(\xi)$ is not parameterized by a neural network, how does the model ensure that the learned sampler contributes beyond a standard MetaD bias? Could the authors clarify whether WT-ASBS should be interpreted as (a) a neural sampler guided by an analytic bias, or (b) a learnable bias parameterized through the neural transport model?
>
> The well-tempered bias $V_{WT}(\xi)$ is intentionally non-learnable, and this design does not conflict with having a learned sampler. The two components serve distinct and complementary purposes. The role of $V_{WT}$ is to provide a low-dimensional exploration signal along a chosen CV space. This is analogous to standard WTMetaD, where the bias controls exploration but does not define the full high-dimensional distribution. In WT-ASBS, the neural sampler (ASBS) is responsible for learning the full Boltzmann distribution over atomic coordinates under the current biased potential $E + V_{WT} \circ \xi$. Only the low-dimensional part of the energy landscape is shaped analytically, and the high-dimensional distribution remains fully learned. This separation is not a limitation but a useful structural choice. A neural network does not replace the physics of bias deposition, and a handcrafted bias cannot reproduce the high-dimensional conditional structure ASBS captures. WT-ASBS should therefore be interpreted as (a) a neural sampler guided by an analytic bias. The bias controls where exploration is encouraged, and the ASBS model learns the full conditional geometry of the molecular system.
>
> > **W2. Lack of fair and direct comparison with ASBS.** Since WT-ASBS is an ASBS variant augmented with a well-tempered bias, the most meaningful benchmark is ASBS itself under identical conditions. However, ASBS only appears in Fig. 3c (explored states) and is not evaluated for PMF accuracy, making the incremental benefit of WT unclear.
> >
> > **W4. Limited diffusion-based baselines.** Comparisons are restricted to WTMetaD, which mainly highlights conceptual differences rather than validating the WT mechanism across diffusion methods. Adding at least one diffusion-based baseline (annealed, energy-guided, or score-guided) would strengthen the evaluation.
>
> A direct comparison with baseline ASBS under identical conditions is indeed essential, and this comparison is provided in Appendix F.1, where we evaluate ASBS and WT-ASBS on alanine dipeptide using standard distributional metrics: KL divergence for 1D torsion marginals, and $\mathcal{W}_2$ and JSD for the joint $(\phi,\psi)$ distribution (Table 2 and Fig. 14). The results show that WT-ASBS matches or improves upon ASBS on the 1D KL metrics, achieves similar $\mathcal{W}_2$, and obtains a lower JSD. Beyond these metrics, the Ramachandran plots in Fig. 14 reveal a more important qualitative distinction. The plotted log densities correspond directly to the negative PMF on $(\phi, \psi)$, which the reviewer specifically asked about. WT-ASBS reconstructs this densities (negative PMF) more faithfully, while ASBS tends to either overconcentrate in the dominant basins or leak into regions with negligible reference probability. This reflects the fact WT-ASBS uses bias-corrected sampling that preserves the correct relative Boltzmann weights across modes.
>
> > **W3. # of force-calls inconsistency between figures.** Figure 3.c shows up to 2 M evaluations, while Figure 3.d extends to 40 M. Because WT-ASBS reaches chemical accuracy after 4–5 M evaluations, truncating Fig. 3c prevents fair comparison at later stages.
> >
> > **Q3.** Could you extend Figure 3.c to the same energy-evaluation range (10–40 M) as Figure 3.d to show late-stage exploration and better quantify the WT bias’s long-term effects? Please include an ASBS PMF accuracy curve for Ala2 system.

---

> > ### Author Response · Authors · 2025-11-25
> > **Replies to Reviewer 2tNT (Part 2)**
> >
> > Fig. 3c is intentionally truncated to the early part of training to highlight the initial exploration phase. In biased sampling methods such as WT-ASBS and WTMetaD, the available metastable states are typically discovered early, and once each state has been sampled sufficiently, the free energies reconstructed from the deposited bias converge during the later stages. Because MD sampling is sequential, the state counts in Fig. 3c accumulate from the start, and by about 2M energy evaluations, all metastable states have already been explored for the biased methods, so extending the curve to 10–40M would not change their exploration statistics. In contrast, Fig. 3d focuses on long-term convergence of the reconstructed PMF, which is why it is plotted over the entire training/sampling range. The two panels therefore use different ranges to highlight different aspects of training: early exploration in Fig. 3c and late-stage PMF accuracy in Fig. 3d.
> >
> > > **W5. Scalability and consistency.** WT-ASBS excels on alanine dipeptide (Figure 2.f) but underperforms WTMetaD on tetrapeptide (Figure 3.d), suggesting potential sensitivity to CV dimensionality or generalizability to complex system.
> >
> > To reproduce free energy differences accurately, both broad exploration across modes and thorough local sampling within each mode matter. Diffusion samplers are effective at generating diverse configurations globally, but MD-based local exploration can be advantageous when the system has many degrees of freedom and intra-mode motion involves barriers that are not very large. In such cases, the sampler must learn all degrees of freedom, which can reduce efficiency and accuracy. We added a note in Discussion section that, for scalable sampling in chemical settings, it is important to determine where global sampling (via measure transport) is needed to overcome significant energetic barriers, and where local sampling is more suitable because movement on the energy landscape is already reasonably efficient. This would also be relevant for generalization to complex systems, such as protein solvated in water. We have incorporated this discussion in the revised manuscript (Section 4.1, p. 8--9).
> >
> > > **W6. Lack of statistical robustness.** The reported results appear to be based on a single simulation run for WT-ASBS. Given the stochastic nature of both diffusion and metadynamics sampling, multiple independent runs (with mean and variance) are crucial to assess convergence stability and reproducibility. Without them, it is difficult to judge whether the reported trajectories and PMF curves reflect consistent behavior or a favorable random seed.
> > >
> > > **Q7.** Are all reported curves from single runs, or averaged over multiple simulations? If single, could you comment on the run-to-run variability? Showing variance bands or repeated trials would help establish statistical reliability.
> >
> > The reported free energy convergence curves for WT-ASBS are from single runs, and the corresponding WTMetaD results are averaged or taken from multiple runs using the same total GPU budget. However, the ablation studies in Appendix E.1 show that WT-ASBS exhibits very similar convergence behavior across different hyperparameter choices, with only modest and predictable changes in convergence speed (Fig. 7). This indicates that run-to-run variability is small compared to the differences between methods. While we focused on using available compute for ablations during the rebuttal period, we will expand the statistical analysis and update the camera-ready version accordingly.
> >
> > > **Q2.** In Figure 8, the reweighted and bias-derived PMFs are nearly identical, suggesting that the sampling in CV space is almost uniform. If so, what ensures that the sampled structures remain physically valid? Could the authors provide representative molecular structures sampled from different regions of the CV space (e.g., high- vs low-free-energy regions) to illustrate what kind of configurations the model actually produces? This would clarify whether the method truly explores meaningful conformations or simply performs uniform random exploration over CVs.
> >
> > While Fig. 15 (Fig. 8 before revision) shows that the PMFs from reweighted samples and from the bias are nearly identical, this does not imply that sampling in the CV space is uniform. The PMF is reconstructed from the bias via $F(s) = -\frac{\gamma}{\gamma - 1} V(s)$, whereas the bias-derived log-weights assigned to samples are $\beta V(s)$. Thus, agreement in PMF indicates that the empirical CV marginal is close to the WT target $\bar{\nu}_{WT}(s) \propto \exp(-\frac{\beta}{\gamma} F(s))$, rather than uniform over the CV domain.

---

> > > ### Author Response · Authors · 2025-11-25
> > > **Replies to Reviewer 2tNT (Part 3)**
> > >
> > > The physical validity of the sampled structures is ensured because the orthogonal degrees of freedom remain unbiased, and although the CV marginal is modified, high-PMF regions (e.g., torsions that cause steric clashes) are still disfavored. The WT target is a tempered distribution with an elevated effective temperature, not a uniform distribution, so unphysical regions are naturally undersampled. In practice, the sampler also receives strong gradient signals from the energy function during training that push it away from high-energy configurations, which preserves physical plausibility. As requested by the reviewer, representative structures from WT-ASBS for alanine tetrapeptide are also provided in the Supplementary Material file for inspection (`ala4_samples/ala4_state_{0..7}.pdb`).
> > >
> > > > **Q4.** Could you summarize cost–benefit metrics (energy evaluations, wall-clock time, ESS, PMF error) across ASBS, WT-ASBS, and WTMetaD for standardized comparison?
> > >
> > > While each experiment highlights different aspects of the cost–benefit trade-offs, providing a single unified summary is difficult because the methods differ fundamentally in how sampling is performed. For example, WT-ASBS and WTMetaD allow on-the-fly estimation of free-energy errors from the current bias, whereas ASBS would require extensive additional sampling to obtain comparable estimates. Instead, we summarize below how each of the suggested metrics is addressed:
> > >
> > > - Wall-clock time: For systems with classical potentials, diffusion-based samplers are inherently slower because neural network inference dominates the cost, whereas MD propagation is inexpensive. For MLIP-based reactive landscapes, Fig. 5 compares wall-clock time versus PMF error for WT-ASBS and WTMetaD (this comparison cannot include ASBS because it cannot sample the required high-energy regions and does not offer on-the-fly PMF estimate).
> > > - Energy evaluations: For $\Delta F$ values reconstructed from the bias (available for ASBS and WT-ASBS), Figs. 2 and 3 report $\Delta F$ as a function of the number of energy evaluations.
> > > - Distributional metrics for generated samples: ESS is not straightforward to compute for diffusion samplers without exact likelihoods, and path importance weights (path ESS) suffer from high variance. Instead, Table 2 and Fig. 14 compare the distributional statistics of ASBS and WT-ASBS based on generated samples.
> > >
> > > > **Q5.** ASBS’s slow exploration (Figure 3.c) might result from strong pre-training confinement near the initial mode (marked * in Figure 3.b). Have you explored reducing pretraining strength or fine-tuning its duration to encourage broader exploration?
> > >
> > > While pretraining does confine the initial distribution near the starting mode, we applied the same pretraining to both ASBS and WT-ASBS because it helps avoid spending many energy evaluations correcting other degrees of freedom in the early stages, where numerous high-energy local minima can trap the sampler. For instance, we observed that ASBS initially generates mainly cis peptide bonds ($\omega$ torsions) for both alanine dipeptide and tetrapeptide, and only transitions to the expected trans configurations after several million energy evaluations. Because the isomerization barrier is large, it is more effective to learn this behavior from the MD-based pretraining distribution than to rely on exploration by the sampler alone. Hence, starting from scratch without pretraining does explore different modes, but it also becomes stuck in unintended local minima, requiring a large number of energy evaluations to reach physically meaningful configurations. This behavior is observed in the pretraining ablation in Appendix E.2 (Fig. 13) for alanine dipeptide, where the torsional distributions of early samples and accumulated bias differ markedly from the expected Ramachandran plot patterns.
> > >
> > > > **Q6.** When computing the free-energy difference, which states or basins were used? Were they predefined minima or clusters in CV space? Clarifying this would help interpret the reported PMF results.
> > >
> > > We thank the reviewer for pointing this out. For the experiments reporting free-energy differences (alanine dipeptide and tetrapeptide), the states were defined using the $\phi$ torsion angles, as indicated by the gray dotted boxes in Fig. 2 (Ala2) and Fig. 15 (Ala4). The exact state definitions were missing in the original submission, and they have now been added to the captions of Fig. 2 (Ala2) and Fig. 3 (Ala4).
> > >
> > > ---
> > >
> > > Thank you,
> > >
> > > Submission13338 authors

---

### Official Review · Reviewer_5dYX · 2025-10-31

**Soundness:** 3
**Presentation:** 3
**Contribution:** 3
**Rating:** 8
**Confidence:** 2

**Summary:**

This paper introduces the Well-Tempered Adjoint Schrödinger Bridge Sampler (WT-ASBS), a novel method that significantly enhances the exploration capabilities of energy-based diffusion models for molecular systems. The core innovation is integrating an adaptive bias, inspired by well-tempered metadynamics (WTMetaD), into the Adjoint Schrödinger Bridge Sampler (ASBS) training loop. This bias is deposited along pre-defined Collective Variables (CVs), which are low-dimensional projections of molecular coordinates. The bias effectively increases the sampling temperature in the CV space, mitigating the mode collapse pitfall often seen in standard diffusion samplers. Crucially, the method retains the ability to recover the correct Boltzmann ensemble through importance reweighting. The authors demonstrate WT-ASBS's efficacy on conformational sampling benchmarks (alanine dipeptide and tetrapeptide) and, notably, achieve the first demonstration of reactive sampling using a diffusion-based model on $S_N2$ and post-transition-state bifurcation reactions, showing significant efficiency gains over traditional WTMetaD in terms of wall-clock time and energy evaluations.

**Strengths:**

The reviewer here acknowledges that I am familiar with MCMC methods and general energy-based sampling, but is less knowledgeable about the specifics of modern molecular sciences applications.

- The work presents a highly original and timely unification of two distinct fields: enhanced sampling (metadynamics) and generative diffusion modeling. By incorporating the well-tempered biasing mechanism directly into the iterative proportional fitting (IPF) scheme of the ASBS, the authors solve the critical problem of mode collapse for diffusion models applied to complex, high-dimensional, and multi-modal free energy landscapes. The successful application of reactive sampling, which is extremely challenging for standard diffusion models, is a significant first.

- The paper is technically sound and well-executed.

  - The authors provide a convergence guarantee (Proposition 3.1), showing that WT-ASBS provably approaches the desired well-tempered target distribution.

  - The experiments are convincing. On the alanine tetrapeptide, WT-ASBS discovers all eight metastable states much faster than WTMetaD (Figure 3c). The ability to accurately resolve complex free energy landscapes (PMFs) for both conformational and reactive systems, with close agreement between PMF from bias and PMF from reweighting, strongly supports the method's correctness and efficiency.

  - For the complex reactive systems using universal ML interatomic potentials (uMLIPs), WT-ASBS achieves convergence with significantly fewer energy evaluations and less wall-clock time compared to WTMetaD (Figure 5).

- The paper is well-structured and clearly written, making complex concepts accessible. The methodological details, especially the two-time-scale algorithm and the practical implementation with a replay buffer (Algorithm 2), are laid out logically. Figure 1 clearly illustrates the overall scheme, and the experimental figures (e.g., Figure 2) effectively show the evolution of the PMF during training.

**Weaknesses:**

The effectiveness of WT-ASBS hinges on the accurate selection of several critical hyperparameters inherited from the enhanced sampling domain, such as the initial Gaussian height $h$, the Gaussian width $\sigma$, and the bias factor $\gamma$.

It seems that the paper lacks a systematic ablation study to investigate the sensitivity of the method to these parameters (e.g., how the convergence speed or final accuracy changes with different $\gamma$ values). Given that WT-ASBS generates uncorrelated samples, the optimal choice for parameters like $h$ might deviate significantly from WTMetaD conventions. It requires clearer guidelines for practitioners.

**Questions:**

See weakness

---

> ### Author Response · Authors · 2025-11-25
> **Replies to Reviewer 5dYX**
>
> We thank the reviewer for the positive feedback and for taking the time to review our work. Please find our point-to-point replies below.
>
> > **W1.** The effectiveness of WT-ASBS hinges on the accurate selection of several critical hyperparameters inherited from the enhanced sampling domain, such as the initial Gaussian height $h$, the Gaussian width $\sigma$, and the bias factor $\gamma$.
> It seems that the paper lacks a systematic ablation study to investigate the sensitivity of the method to these parameters (e.g., how the convergence speed or final accuracy changes with different $\gamma$ values). Given that WT-ASBS generates uncorrelated samples, the optimal choice for parameters like $h$ might deviate significantly from WTMetaD conventions. It requires clearer guidelines for practitioners.
>
> To address the sensitivity of WT-ASBS to the WT bias hyperparameters, we added a systematic ablation study covering $h$, $\sigma$, and $\gamma$ (Appendix E.1, Figs. 7 through 10). Across a wide range of settings, the method shows stable convergence of the free energy difference $\Delta F$. For the Gaussian height, values from 1.5e-5 to 5e-4 eV produce nearly identical final $\Delta F$, even though they differ by almost a factor of 30. Larger $h$ accelerates early exploration but can slightly reduce final accuracy only at the highest tested value, 1.5e-3 eV. For $\sigma$, broader kernels promote faster initial spreading of the bias but have limited influence on the final PMF reconstruction. For $\gamma$, small values slow exploration and delay convergence, while moderate values lead to fast and consistent results. Taken together, these experiments provide practical guidance and show that WT-ASBS remains effective without careful tuning of these parameters.
>
> In addition to the hyperparameter study, we also examined two training mechanism components that, although not requested, directly influence efficiency (Appendix E.2). First, the replay buffer ablation (Fig. 11) shows that too few ASBS updates between buffer refreshes slow convergence because the sampler cannot keep up with the evolving bias. Second, warm starting through localized pretraining yields faster early progress and reduces the number of energy evaluations, as shown in Figs. 11 and 13. These results clarify how the practical implementation aligns with the ideal two-time-scale formulation of WT-ASBS.
>
> ---
>
> Thank you,
>
> Submission13338 authors

---

### Official Review · Reviewer_xH8L · 2025-11-03

**Soundness:** 3
**Presentation:** 3
**Contribution:** 3
**Rating:** 6
**Confidence:** 4

**Summary:**

In this work, the authors present a collective-variable-based approach to enhance diffusion-based samplers. By incorporating a biasing force that is gradually deposited, similar to metadynamics, the approach improves the exploration ability of diffusion-based samplers. This addresses a consistent issue with diffusion-based samplers being susceptible to mode collapse.

The specific diffusion-based sampler enhanced using collective variables (CVs) is the Adjoint Schrödinger Bridge Sampler (ASBS), although the CV-based approach does not appear to be necessarily specific to this instance of diffusion-based sampler. The biasing along the CV is approached similarly to most biased enhanced-sampling methods: an additional bias potential is added to the base potential energy. This bias potential is then updated using the same rule as in well-tempered MetaD. This results in a two-step optimization process: an inner step, in which ASBS is performed (resulting in mode collapse), and an outer step, in which the bias potential is updated. In addition, a few practical improvements are implemented, such as the use of a replay buffer and a warm start.

The presented method is evaluated on a number of interesting systems. First, the evaluation considers two peptides, Alanine Dipeptide and Alanine Tetrapeptide, and, following this, the authors study reactive pathways in the form of a simpler SN2 reaction and a more complex reaction with multiple products.

**Strengths:**

The authors present an interesting and seemingly successful approach to alleviate the well-known issue of mode collapse in diffusion-based samplers within the domain of molecular conformations. The focus on ASBS as the diffusion-based sampler to enhance, and the choice of the well-tempered bias update, also seems the most sensible approach. Furthermore, the paper is well written and does an excellent job highlighting the core considerations that go into an enhanced-sampling method.

The experimental evaluation is relatively in-depth and focuses on two interesting problems by considering both simple peptides and reactions. While quantitative results are limited, the presented results suggest a significant improvement over standard WTMetaD.

All in all, the paper presents an excellent approach to enhance diffusion-based samplers to make them more appropriate for the equilibrium sampling of a molecular system. As such, I vote to accept the paper for publication.

**Weaknesses:**

- Looking at Figure 2, I’m surprised to see roughly uniform sampling from the second outer-loop iteration onwards (assuming that this is what the dots in panel c represent). I would have expected the samples to be located around the two smaller modes at this point, or still located at the original modes. Only at the later iterations would I expect to see roughly uniform sampling. Notably, this is also directly in contrast with the intuition presented in Figure 1. An author response to this inconsistency is required for me to increase my score beyond a borderline accept.
- There are a number of important components to the training setup that would be good to see studied more in-depth, such as the use of a replay buffer and the warm start. Most importantly, I am interested in seeing how these components influence the number of energy evaluations needed.
- Due to the reliance on collective variables, it is difficult to envision the applicability of the presented approach outside domains where CVs are known and provide a clear boundary to the sampling domain. For example, in areas such as the study of protein folding dynamics, it is hard to define CVs that sufficiently restrict the important sampling domain.
- As the work could potentially spark new interest in enhanced sampling, it would be good to see a more in-depth discussion of work in this area, such as umbrella sampling and adaptive biasing force methods. This could potentially be combined with the discussion of the requirements in Section 2.1, whose current purpose is not entirely clear. Additionally, it would be good to see additional works discussed, such as [1–4] (not a complete list), all of which use a similar biasing force to that proposed here in the context of enhanced sampling (but are more focused on transition path sampling).

[1] Seong, Kiyoung, et al. “Transition Path Sampling with Improved Off-Policy Training of Diffusion Path Samplers.” arXiv preprint arXiv:2405.19961 (2024).
[2] Singh, Aditya N., Avishek Das, and David T. Limmer. “Variational path sampling of rare dynamical events.” Annual Review of Physical Chemistry 76 (2025).
[3] Holdijk, Lars, et al. “Stochastic optimal control for collective-variable-free sampling of molecular transition paths.” Advances in Neural Information Processing Systems 36 (2023): 79540–79556.
[4] Du, Yuanqi, et al. “Doob’s Lagrangian: A Sample-Efficient Variational Approach to Transition Path Sampling.” Advances in Neural Information Processing Systems 37 (2024): 65791–65822.

**Questions:**

See weaknesses.

**Details Of Ethics Concerns:**

-

---

> ### Author Response · Authors · 2025-11-25
> **Replies to Reviewer xH8L (Part 1)**
>
> We thank the reviewer for the constructive and supportive comments and for carefully reviewing our work. Please find our point-to-point replies below:
>
> > **W1.** Looking at Figure 2, I’m surprised to see roughly uniform sampling from the second outer-loop iteration onwards (assuming that this is what the dots in panel c represent). I would have expected the samples to be located around the two smaller modes at this point, or still located at the original modes. Only at the later iterations would I expect to see roughly uniform sampling. Notably, this is also directly in contrast with the intuition presented in Figure 1. An author response to this inconsistency is required for me to increase my score beyond a borderline accept.
>
> We agree that this apparent deviation from the ideal WT-ASBS behavior needs further clarification. The effect comes from two factors: (1) how the PMF is visualized in Fig. 2 and (2) the difference between the two-time-scale WT-ASBS setup where the inner-loop ASBS training is run to convergence (Algorithm 1, which produces the behavior in Fig. 1) and the practical implementation (Algorithm 2) where the bias and sampler parameters are updated at the same time.
>
> First, the PMF in Fig. 2 is truncated, so high PMF regions that correspond to areas with small deposited bias are not shown. To give a more complete picture, we added in Appendix E.1 (Fig. 8) a scatter plot of samples with the deposited bias visualized. Although Fig. 2 may make the sample exploration look much more earlier than the bias/PMF update, in the practical setting the sampler parameters and the bias evolve almost concurrently.
>
> However, the reviewer is correct that at intermediate training steps the sampler produces samples in CV regions that would be less populated under the Boltzmann distribution defined by the current energy + bias. Fig. 1 illustrates the idealized case where the sampler is trained to convergence between bias updates (Algorithm 1). In contrast, to reduce the number of energy evaluations, the practical implementation (Algorithm 2) uses a replay buffer with a fixed number of gradient updates between bias updates, and the updated samples directly populate the replay buffer. This produces a temporary deviation of the ASBS control model from the equilibrium Boltzmann distribution at the current bias in the early stage of training. The deviation becomes smaller over time because continued training and the exponential decay in the WT bias update slow the bias accumulation, eventually giving converged bias and sampler parameters.
>
> This effect is visible in the added ablation results for the Gaussian height $h$ in Fig. 8. When $h$ is small, the CV distribution of samples stays concentrated near the modes (for example, $h$ 1.5e-5, 1.66M E evals). When $h$ is large, the CV distribution becomes broader (for example, $h$ 5e-4 and 1.5e-3, 0.35M E evals). A larger $h$ initially pushes the sampler away from equilibrium, with the added noise in the diffusion process that spreads the distribution. However, the bias acts only in CV space, so large off-equilibrium behavior appears in the CVs while the orthogonal degrees of freedom (bond lengths, angles, and other torsions) remain mostly unchanged, which keeps the sampled configurations physically valid.
>
> > **W2.** There are a number of important components to the training setup that would be good to see studied more in-depth, such as the use of a replay buffer and the warm start. Most importantly, I am interested in seeing how these components influence the number of energy evaluations needed.
>
> We have added new ablation experiments that examine how the replay buffer and warm start influence efficiency, measured by the number of energy evaluations required for convergence.
>
> The replay buffer ablation in Fig. 11 shows that using only a small number of ASBS updates between buffer refreshes (reducing the amount of replay buffer use) slows convergence considerably. The bias deposition patterns in Fig. 12 confirm the cause: when buffer and bias updates occur too frequently, the sampler cannot keep up with the evolving bias, which leads to excessive bias deposition in the initial basin and delays exploration. Increasing the number of ASBS updates between buffer updates brings the behavior closer to the ideal two-time-scale scheme and reduces the number of energy evaluations needed.
>
> Fig. 11 also shows that warm starting through localized pretraining leads to much faster convergence. As illustrated in Fig. 13, without pretraining, the model must identify low-energy modes from scratch and requires many more energy evaluations in the early stages. Pretraining provides a good initialization that makes the joint evolution of sampler and bias more efficient.

---

> ### Author Response · Authors · 2025-11-25
> **Replies to Reviewer xH8L (Part 2)**
>
> In addition to what the reviewer asked, we also added ablations of the well-tempered bias hyperparameters. These results (Figs. 9 and 10) show that WT-ASBS is robust across broad ranges of $h$, $\sigma$, and $\gamma$, and that moderate variations in these parameters do not make a large difference in the number of energy evaluations required for convergence.
>
> > **W3.** Due to the reliance on collective variables, it is difficult to envision the applicability of the presented approach outside domains where CVs are known and provide a clear boundary to the sampling domain. For example, in areas such as the study of protein folding dynamics, it is hard to define CVs that sufficiently restrict the important sampling domain.
>
> The reviewer is correct that the method needs a bounded, low-dimensional CV space to place bias. Our focus in this paper is on validating WT-ASBS in energy-based training settings, so we primarily used benchmark systems with well-defined equilibrium distributions with known collective variables, while larger systems such as protein folding/unfolding with large degrees of freedom is outside our scope here.
>
> That said, the approach is not limited to known hand-crafted CVs. A practical approach to complex systems is first to identify metastable modes via data-based generative models, then learn a CV that separates those modes, and run WT-ASBS using the learned CV. To support this, we added an ML CV experiment for alanine dipeptide (Appendix F.4) where a one-dimensional Deep-TDA CV is learned from unbiased state samples. WT-ASBS on this learned CV recovers the correct 1-D PMF, gives smooth convergence of $\Delta F$, and yields a reweighted 2-D PMF in good agreement with reference data. Alongside the hand-designed CV results, this shows that WT-ASBS works with both standard CVs and automatically generated CVs, and points to a possible extension route for higher-dimensional systems.
>
> > **W4.** As the work could potentially spark new interest in enhanced sampling, it would be good to see a more in-depth discussion of work in this area, such as umbrella sampling and adaptive biasing force methods. This could potentially be combined with the discussion of the requirements in Section 2.1, whose current purpose is not entirely clear. Additionally, it would be good to see additional works discussed, such as [1–4] (not a complete list), all of which use a similar biasing force to that proposed here in the context of enhanced sampling (but are more focused on transition path sampling).
> [1] Seong, Kiyoung, et al. “Transition Path Sampling with Improved Off-Policy Training of Diffusion Path Samplers.” arXiv preprint arXiv:2405.19961 (2024). [2] Singh, Aditya N., Avishek Das, and David T. Limmer. “Variational path sampling of rare dynamical events.” Annual Review of Physical Chemistry 76 (2025). [3] Holdijk, Lars, et al. “Stochastic optimal control for collective-variable-free sampling of molecular transition paths.” Advances in Neural Information Processing Systems 36 (2023): 79540–79556. [4] Du, Yuanqi, et al. “Doob’s Lagrangian: A Sample-Efficient Variational Approach to Transition Path Sampling.” Advances in Neural Information Processing Systems 37 (2024): 65791–65822.
>
> We appreciate the reviewer’s suggestions, and we have included a discussion of different enhanced-sampling variants in Appendix A.1 and referenced it in Section 2.1. We also cited and discussed the suggested works on ML-based TPS methods in Appendix A.3 for extended discussion on related works.
>
> ---
>
> Thank you,
>
> Submission13338 authors

---

### Author Response · Authors · 2025-12-02
**Summary of the Discussion Period**

We would like to thank all reviewers for their constructive reviews and engagement during the discussion period. Here, we summarize the highlighted strengths and the revisions/clarifications made during the discussion.

### Strengths

- All reviewers unanimously highlighted the novelty and the performance of the proposed method.
  - The integration of diffusion sampler with well-tempered bias update was seen as "excellent approach to enhance diffusion-based samplers" (**xH8L**), "a highly original and timely unification of two distinct fields" (**5dYX**), “conceptually innovative hybridization” (**2tNT**), and "elegant and promising" (**mrdW**).
- The quality and clarity of the manuscript were reflected in its favorable initial ratings (**6/8/6/6**), as well as the reviewer comments.
  - "the paper [...] does an excellent job highlighting the core considerations" (**xH8L**), “technically sound and well-executed” (**5dYX**), and “clearly written, logically organized, and visually well-supported by figures” (**mrdW**).

### Common revisions and clarifications

- Reviewers **xH8L** and **2tNT** commented on the apparent early "uniform" sampling and the physical validity of sampled structures.
  - We added scatter plots of samples with deposited bias and ablation studies in Appendix E.1, clarified the difference between the ideal two-time-scale WT-ASBS and the practical concurrent-update implementation, and explained why broadened CV marginals do not imply uniform sampling and still yield physically valid configurations.
- Reviewers **xH8L** and **5dYX** requested more systematic studies of replay buffer, warm start, and well-tempered hyperparameters and their impact on efficiency.
  - We added ablations over Gaussian height, width, and bias factor, plus replay-buffer and warm-start settings in Appendices E.1 and E.2, reported convergence behavior and required energy evaluations, and distilled practical guidelines for choosing these parameters (Appendix C.4).
- Reviewers **xH8L** and **mrdW** raised concerns about reliance on chosen predefined CVs, which may hinder the broader applicability of the method.
  - We added a WT-ASBS experiment with a learned ML CV (Deep-TDA) in Appendix F.4, showing that WT-ASBS works with automatically learned CVs and can serve as a downstream sampler once informative low-dimensional coordinates are available.

---

### Meta-Review · Area_Chair_nwbT · 2025-12-19

**Summary:**

This paper presents a compelling integration of diffusion-based generative models with well-tempered metadynamics (WT-ASBS), addressing a fundamental hurdle in molecular simulation: the tendency for diffusion samplers to collapse into local energy minima. By using low-dimensional collective variables to drive exploration, the authors bridge the gap between generative AI and established physical sampling paradigms. This is not just a theoretical exercise; the work marks the first successful use of diffusion models to capture reactive energy landscapes—specifically bond breaking and formation—with an efficiency that rivals or exceeds traditional molecular dynamics.

The discussion phase was particularly productive in resolving initial skepticism. While reviewers were concerned about the reliance on handcrafted collective variables, the authors provided convincing evidence that the framework works just as well with machine-learned CVs. This significantly broadens the method's applicability to complex systems where reaction coordinates aren't known a priori. Furthermore, the rebuttal clarified that the sampler remains physically grounded even under heavy bias, and that the performance is robust across various hyperparameter settings.

While some might argue the innovation leans toward system integration, the contribution to "AI for Science" is undeniable. Providing a rigorous, efficient way to solve mode-collapse in molecular sampling is a major methodological win. The authors have demonstrated a high level of technical maturity, and the resulting framework is a significant step toward practical, near-first-principles molecular modeling.

The paper represents a significant technical contribution by being the first to demonstrate reactive sampling with near-first-principles accuracy using a diffusion sampler. It achieves a substantial reduction in wall-clock time compared to traditional WTMetaD for complex chemical reactions (e.g., SN2). The authors successfully addressed the most critical technical critiques (CV dependency and hyperparameter sensitivity) during the rebuttal. However, the slightly lower accuracy than MD on larger systems (tetrapeptide) and the reliance on single-run statistics for some results suggest it is better suited as a poster paper.

**Reviewer Concerns:**

Addressed by Rebuttal

- Hyperparameter Robustness: The authors added systematic ablation studies in Appendix E.1. They demonstrated that WT-ASBS is stable across a 30-fold range of $h$ values and provided practical order-of-magnitude guidelines
- Predefined CV Dependency: To address concerns about broader applicability, the authors performed a new experiment using a learned ML CV (Deep-TDA). The results in Appendix F.4 show the method successfully reconstructs PMFs and free energy differences using automatically discovered coordinates.
- Training Mechanisms: The authors clarified the role of the replay buffer and pretraining in Appendix E.2. They showed that localized pretraining significantly reduces the energy evaluations needed to identify low-energy modes.
- Early Sampling/Physicality: Authors clarified that while CV marginals may appear broader (approaching the well-tempered target, not true uniformity), orthogonal degrees of freedom (bonds, angles) remain unbiased and physically valid due to the underlying energy gradient signals.

Outstanding Concerns

- Statistical Reliability: While authors provided extensive ablations, some reviewers noted that the main convergence curves for WT-ASBS were from single runs. Although authors argue variability is small based on ablations, multiple independent trials for the primary benchmarks were not fully integrated into the revised draft during the rebuttal.
- Comparison at Scale: The method underperformed WTMetaD on the alanine tetrapeptide in terms of final free energy accuracy. While authors provided a valid scientific reason (MD’s advantage in local intra-mode mixing for high-degree-of-freedom systems), this defines a performance boundary for the current version of the algorithm.

**Reviewer Scores:**

- xH8L (possibly from score 6 to score 8): Explicitly stated response was "required for me to increase my score". Rebuttal provided the requested ablations and ML-CV experiments.
- 5dYX (score 8): Highly positive initially; rebuttal addressed their primary concern regarding hyperparameter guidelines.
- 2tNT (score 6): Rebuttal clarified the "conceptual tension" regarding the non-neural bias and provided representative structures to prove physical validity.
- mrdW (score 6): Remained positive but maintained current score. Still has slight reservations about ML methodology vs. application focus.

---

### Decision · Program_Chairs · 2026-01-26

Accept (Poster)